# The Surprising Power of Graph Neural Networks with Random Node Initialization

## Abstract

Graph neural networks (GNNs) are effective models for representation learning on graph-structured data. However, standard GNNs are limited in their expressive power, as they cannot distinguish graphs beyond the capability of the Weisfeiler-Leman (1-WL) graph isomorphism heuristic. This limitation motivated a large body of work, including higher-order GNNs, which are provably more powerful models. To date, higher-order invariant and equivariant networks are the only models with known universality results, but these results are practically hindered by prohibitive computational complexity. Thus, despite their limitations, standard GNNs are commonly used, due to their strong practical performance. In practice, GNNs have shown a promising performance when enhanced with *random node initialization (RNI)*, where the idea is to train and run the models with randomized initial node features. In this paper, we analyze the expressive power of GNNs with RNI, and pose the following question: are GNNs with RNI more expressive than GNNs? We prove that this is indeed the case, by showing that GNNs with RNI are *universal*, a first such result for GNNs not relying on computationally demanding higher-order properties. We then empirically analyze the effect of RNI on GNNs, based on carefully constructed datasets. Our empirical findings support the superior performance of GNNs with RNI over standard GNNs. In fact, we demonstrate that the performance of GNNs with RNI is often comparable with or better than that of higher-order GNNs, while keeping the much lower memory requirements of standard GNNs. However, this improvement typically comes at the cost of slower model convergence. Somewhat surprisingly, we found that the convergence rate and the accuracy of the models can be improved by using only a *partial* random initialization regime.

## 1 Introduction

Graph neural networks (GNNs) (Scarselli et al., 2009; Gori et al., 2005) are neural architectures designed for learning functions over graph-structured data, and naturally encode desirable properties such as permutation invariance (resp., equivariance) relative to graph nodes, and node-level computation based on message passing between these nodes. These properties provide GNNs with a strong inductive bias, enabling them to effectively learn and combine both local and global graph features (Battaglia et al., 2018). As a result, GNNs have been applied to a multitude of tasks, ranging from protein classification (Gilmer et al., 2017) and synthesis (You et al., 2018), protein-protein interaction (Fout et al., 2017), and social network analysis (Hamilton et al., 2017), to recommender systems (Ying et al., 2018) and combinatorial optimization (Bengio et al., 2018; Selsam et al., 2019).

However, popular GNN architectures, primarily based on message passing (MPNNs), are limited in their expressive power. In particular, MPNNs are at most as powerful as the Weisfeiler-Leman (1-WL) graph isomorphism heuristic (Morris et al., 2019; Xu et al., 2019), and thus cannot discern between several families of non-isomorphic graphs, e.g., sets of regular graphs (Cai et al., 1992). To address this limitation, alternative GNN architectures with provably higher expressive power than MPNNs have been proposed. These models, which we refer to as *higher-order GNNs*, are inspired by the more powerful generalization of 1-WL to $k$−tuples of nodes, known as $k$-WL (Grohe, 2017). These models are the only GNNs with an established universality result, but these models are computationally very demanding. As a result, MPNNs, despite their limited expressiveness, remain the standard GNN model for graph learning applications.

In a parallel development, MPNNs have recently achieved significant empirical improvements using *random node initialization* (RNI), through which initial graph node embeddings are randomly set. Indeed, RNI has enabled MPNNs to distinguish instances that 1-WL cannot distinguish, and is proven to enable better approximation of a class of combinatorial problems (Sato et al., 2020). However, the effect of RNI on the expressive power of GNNs has not yet been comprehensively studied, and its impact on the inductive capacity and learning ability of GNNs remains unclear.

In this paper, we thoroughly study the impact of RNI on MPNNs. First, we prove that MPNNs enhanced with RNI are universal, in the sense that they can approximate every function defined on graphs of any fixed order. This follows from a logical characterisation of the expressiveness of MPNNs (Barceló et al., 2020) combined with an argument on order-invariant definability. Our result strongly contrasts with existing 1-WL limitations for deterministic MPNNs, and provides a foundation for developing very expressive and memory-efficient MPNN models.

To empirically verify our theoretical findings, we carry out a careful empirical study to quantify the practical impact of RNI. To this end, we design EXP, a synthetic dataset requiring 2-WL expressive power for models to achieve above-random performance, and run MPNNs with RNI on it, to observe *how well* and *how easily* this model can learn and generalize based on this dataset. Then, we propose CEXP, a modification of EXP with partially 1-WL distinguishable data, and evaluate the same questions in this more variable setting. Overall, the contributions of this paper are as follows:

- We prove that MPNNs with RNI are universal, a significant improvement over the 1-WL limit of standard MPNNs and, to our knowledge, a first universality result for memory-efficient GNNs.

- We introduce two carefully designed datasets, EXP and CEXP, based on graph pairs only distinguishable by 2-WL or higher, to rigorously evaluate the impact of RNI.

- Using these datasets, we thoroughly analyze the effects of RNI on MPNN, and observe that (i) MPNNs with RNI can closely match the performance of higher-order GNNs, (ii) the improved performance of MPNNs with RNI comes at the cost of slower convergence (compared to higher-order GNNs), and (iii) using a partial random initialization regime over node features typically improves convergence rate and the accuracy of the models.

- We additionally perform the same experiments with analog, sparser datasets, with longer training, and observe similar behavior, but more volatility.

## 2 GRAPH NEURAL NETWORKS

Graph neural networks (GNNs) (Gori et al., 2005; Scarselli et al., 2009) are neural architectures dedicated to learning functions over graph-structured data. In a GNN, nodes in the input graph are assigned vector representations, which are updated iteratively through series of *invariant* or *equivariant* computational layers. We recall message passing neural networks (MPNNs) (Gilmer et al., 2017), a popular family of GNN models, and its expressive power in relation to the Weisfeiler-Leman graph isomorphism heuristic. We discuss alternative GNN models in Section 3; for a broader coverage, we refer the reader to the literature (Hamilton, 2020).

In MPNNs, node representations aggregate *messages* from their neighboring nodes, and use this information to iteratively update their representations. Formally, given a node $x$, its vector representation $v_{x,t}$ at time $t$, and its neighborhood $N(x)$, a message passing update can be written as:

$$v_{x,t+1} = combine\Big(v_{x,t}, aggregate\big(\{v_{y,t}|\, y \in N(x)\}\big)\Big),$$

where *combine* and *aggregate* are functions, and *aggregate* is typically permutation-invariant. Once message passing is complete, the final node representations are then used to compute target outputs. Prominent message passing GNN architectures include graph convolutional networks (GCNs) (Kipf & Welling, 2017) and gated graph neural networks (GGNNs) (Li et al., 2016).

It is well-known that standard MPNNs have the same power as the 1-dimensional Weisfeiler-Leman algorithm (1-WL) (Xu et al., 2019; Morris et al., 2019). This entails that two nodes in a graph cannot be distinguished if 1-WL does not distinguish them, and neither can two graphs be distinguished if 1-WL cannot distinguish them.

Consider the graphs $G$ and $H$ shown in Figure 1. 1-WL cannot distinguish any two nodes of the graph $G$. Thus, for example, the invariant function $f : V(G) \to \mathbb{R}$ that maps all nodes in the 4-cycle of $G$ to 1 and all nodes in the triangle to 0 is not expressible (or approximable) by a GNN. Moreover, 1-WL cannot distinguish the graphs $G, H$, even though they are obviously non-isomorphic, so no classifier based on the node embeddings computed by an MPNN can distinguish the two graphs.

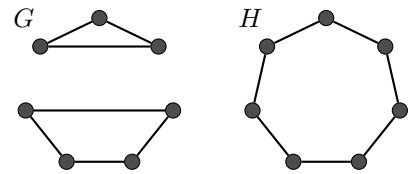

Figure 1: $G$ and $H$ are indistinguishable by 1-WL and hence by (1-WL) GNNs.

A somewhat trivial limitation in the expressiveness of MPNNs is that information is only propagated along edges, and hence can never be shared between distinct connected components of a graph (Barceló et al., 2020; Xu et al., 2019). An easy way to overcome this limitation is by adding *global readouts*, that is, permutation-invariant functions that aggregate the current states of all nodes[1]. Throughout the paper, we therefore focus on MPNNs with global readouts (also called aggregate-combine GNNs with global readout, i.e., *ACR-GNNs* (Barceló et al., 2020)).

## 3 RELATED WORK & MOTIVATION

Developing more expressive GNNs is an active research area due to the prominence of GNNs for relational learning (Hamilton et al., 2017) and combinatorial optimization (Bengio et al., 2018). As mentioned earlier, standard GNN models are at most as expressive as 1-WL (Morris et al., 2019; Xu et al., 2019), and thus cannot distinguish between non-isomorphic input instances. In this section, we describe theoretical results quantifying the expressive power of existing GNNs.

**Higher-order GNNs.** We recall the following families of higher-order GNN models:

- **Invariant (resp., equivariant) graph networks**: Invariant (resp., equivariant) graph networks (Maron et al., 2019b) represent graphs as a tensor where node adjacency is directly encoded, and implicitly pass information between nodes through invariant (resp., equivariant) computational blocks. Hence, these models are themselves invariant (resp., equivariant), a desirable property for computations on graphs. Following intermediate blocks, *higher-order* tensors are typically returned, and the order of these tensors correlates directly with the expressive power of the overall model. Indeed, invariant networks (Maron et al., 2019c), and later equivariant networks (Keriven & Peyré, 2019), are shown to be universal, but with tensor orders of $O(|V|^2)$, where $|V|$ denotes the number of graph nodes. Furthermore, invariant (resp., equivariant) networks with intermediate tensor order $k$ are shown to be equivalent in power to $(k-1)$-WL (Maron et al., 2019a), which is strictly more expressive as $k$ increases (Cai et al., 1992). Therefore, such universal higher-order models require **intractably-sized intermediate tensors** in practice.

- **Higher-order MPNNs**: The $k$−WL hierarchy has been directly emulated in GNNs, such that these models learn embeddings for *tuples* of nodes, and perform messaging passing between them, as opposed to individual nodes. This approach has yielded models such as $k$-GNNs (Morris et al., 2019). $k$-GNNs have $(k-1)$-WL expressive power,[2] but need $O(|V|^k)$ memory to run, leading to **excessive memory requirements**.

- **Provably powerful graph networks (PPGNs)**: PPGN is an invariant GNN (Maron et al., 2019a), based on "blocks" of multilayer perceptrons (MLPs) and matrix multiplication, which theoretically has 2-WL expressive power, and only requires memory $O(|V|^2)$ (compared to $O(|V|^3)$ for 3-GNNs). However, PPGN theoretically requires **exponentially many samples** in the number of graph nodes to learn necessary functions for 2-WL expressiveness (Puny et al., 2020).

**GNNs with random node initialization.** MPNNs have been enhanced with random node initialization (Sato et al., 2020), such that the model trains and runs with partially randomized initial

---

[1]In the terminology of Maron et al. (2019c), such global readouts are tensors of order 1 that are invariant under the symmetric group of the vertex set of the input graph.

[2]In the literature, one can find different (though equally expressive) versions of the Weisfeiler Leman algorithm leading to inconsistent dimension counts. For example, the $(k+1)$-WL and the $(k+1)$-GNNs of Morris et al. (2019) are equivalent in expressiveness to the $k$-WL of Cai et al. (1992); Grohe (2017). We follow the version of Cai et al. (1992), as it has been adopted as the standard in the literature on graph isomorphism testing.

node features. These models, denoted rGNNs, are shown to near-optimally approximate solutions to specific combinatorial optimization problems, and are able to distinguish between 1-WL indistinguishable graph pairs. rGNNs can also detect characteristic sub-graphs in an input graph with high probability. Nonetheless, it remains *open* as to how much expressive power is exactly gained through RNI, and, in general, whether a GNN model that is universal, scalable, and structure-preserving, can be developed. Our work strongly improves the theoretical result of Sato et al. (2020), as it shows *universality* of MPNNs with RNI, and thus that arbitrary real-valued functions over graphs can be learned by MPNNs with the help of RNI. On the empirical side, we highlight the power of RNI in a significantly more challenging setting than rGNN, using a target function (SAT) beyond their theoretical scope. Indeed, for SAT, approximation is known to be *hard*, and fixed local structures are not useful for prediction.

Similar work to RNI has also been conducted in terms of randomly adding features from a pre-determined set of colors (Dasoulas et al., 2020) to disambiguate between nodes. This model, known as CLIP, is similar in spirit to RNI, in that it introduces randomness to node representations, but explicitly makes graphs distinguishable by construction. By contrast, we study random features produced by RNI, which (i) are not designed a priori to distinguish nodes, (ii) do not explicitly introduce a fixed underlying structure, and (iii) yield potentially infinitely many representations for a single graph. In this more general setting, we nonetheless show that RNI adds expressive power to distinguish between nodes with high probability, leads to a universality result, and performs strongly in challenging problem settings.

## 4 RANDOM NODE INITIALIZATION MAKES GNNS UNIVERSAL

We present the main result of the paper, showing that random node initialization significantly increases the expressiveness and makes MPNNs universal, in a natural sense. Our work is a first positive result for the universality of MPNNs. This result is not based on a new model, but rather on random initialization of node features, which is widely used in practice, and in this respect, it also serves as a theoretical justification for models that are successfully employed in practice.

It may appear somewhat surprising, and even counter-intuitive, that randomly initializing node features, on its own, would deliver such a gain in expressiveness. In fact, on the surface, random initialization no longer preserves the invariance of MPNNs, since the result of the computation of an MPNN with RNI not only depends on the structure (i.e., the isomorphism type) of the input graph, but also on the random initialization. The broader picture is, however, rather subtle, as we can view such a model as computing a random variable (or as generating an output distribution), and this random variable would still be invariant. This means that the outcome of the computation of an MPNN with RNI does still *not* depend on the specific representation of the input graph, which fundamentally maintains invariance. Indeed, random features vary around a mean which, in expectation, will inform GNN predictions, and is identical across all nodes as randomization is i.i.d. However, the variability between different samples, and the variability of a random sample relative to this mean, enable graph discrimination and improve expressiveness. Hence, in expectation, all samples over training and evaluation fluctuate around a unique value, preserving invariance, whereas single-sample variance achieves the improved expressiveness.

Formally, let $\mathcal{G}_n$ be the class of all $n$-vertex graphs, i.e., graphs that consist of at most $n$ vertices, and let $f : \mathcal{G}_n \to \mathbb{R}$. We say that $f$ is *invariant* if for isomorphic graphs $G, H \in \mathcal{G}_n$ it holds that $f(G) = f(H)$. We say that a randomized function $\mathcal{X}$ that associates with every graph $G \in \mathcal{G}_n$ a random variable $\mathcal{X}(G)$ is an $(\epsilon, \delta)$-*approximation* of $f$ if for all $G \in \mathcal{G}_n$ it holds that $\Pr(|f(G) - \mathcal{X}(G)| \leq \epsilon) \geq 1 - \delta$. Note that MPNNs with RNI compute functions $\mathcal{X}$ of this type. If $\mathcal{X}$ is computed by an MPNN $\mathcal{N}$ with RNI, we say that $\mathcal{N}$ $(\epsilon, \delta)$-*approximates* $f$.

**Theorem 4.1** (Universal approximation). *Let $n \geq 1$, and let $f : \mathcal{G}_n \to \mathbb{R}$ be invariant. Then, for all $\epsilon, \delta > 0$, there is a MPNN with RNI that $(\epsilon, \delta)$-approximates $f$.*

For ease of presentation, we state the theorem only for real-valued functions, but it can be extended to equivariant functions defined on the nodes of a graph (that is, functions $f$ mapping graphs $G$ to vectors $\boldsymbol{x} \in \mathbb{R}^{V(G)}$ with the property that for every permutation $\pi$ of $V(G)$ it holds that $f(G^\pi) = f(G)^\pi$) and even higher-order equivariant functions. The result can also be extended to weighted graphs, but then the function $f$ that we approximate needs to be continuous.

To prove Theorem 4.1, we first show that MPNNs with RNI can capture arbitrary Boolean functions, by building on the result of Barceló et al. (2020), which states that any logical sentence in $\mathsf{C}^2$ can be captured by an MPNN with global readout. The logic $\mathsf{C}$ is the extension of first-order predicate logic using counting quantifiers of the form $\exists^{\geq k} x$ for $k \geq 0$, where $\exists^{\geq k} x \varphi(x)$ means that there are at least $k$ elements $x$ satisfying $\varphi$, and $\mathsf{C}^2$ is the two-variable fragment of $\mathsf{C}$ (see appendix for further details). We establish that any graph with identifying node features, which we call *individualized graphs*, can be represented by a sentence in $\mathsf{C}^2$. Then, we extend this result to sets of individualized graphs, and thus to Boolean functions mapping these sets to True, by showing that these functions are represented by a $\mathsf{C}^2$ sentence, namely the disjunction of all constituent graph sentences. Following this, we provide a construction with node embeddings based on random node initialization, and show that, with high probability, RNI makes the input graph individualized. Thus, with high probability, RNI makes that MPNNs learn a Boolean function over individualized graphs. Since all such functions can be captured by a sentence in $\mathsf{C}^2$, and an MPNN with a global readout can capture any Boolean function by Barceló et al. (2020), we conclude that MPNNs with RNI can capture arbitrary Boolean functions. Finally, the result is extended to real-valued functions via a natural mapping, yielding universality.

The concrete implications of Theorem 4.1 can be summarized as follows: First, MPNNs enhanced with RNI are able to distinguish individual graphs, already with an embedding dimensionality polynomial in the inverse of desired confidence $\delta$, namely $O(n^2 \delta^{-1})$, where $n$ is the number of graph nodes. Second, our universality results holds also with partial RNI. More specifically, it already holds with only one randomized dimension. Third, although Theorem 4.1 can potentially result in very large constructions, we note that it is very adaptive and tightly linked to the descriptive complexity of the function that we want to approximate. That is, for a more restricted class of functions, there may be more efficient constructions, and our proof does not rely on a particular construction. This deserves a more thorough investigation, which we leave for future work. Finally, our construction provides a *logical characterization* of the power of MPNNs with RNI, and substantiates the means through which randomization yields expressiveness improvements. This construction therefore also serves as a basis for a more logically grounded theoretical study of randomized MPNN models, based on particular architectural or parametric choices.

Similarly to other universality results, Theorem 4.1 can potentially result in very large constructions. This is a simple consequence of the general nature of such universality results: Theorem 4.1 applies to families of functions, describing problems of *arbitrary* computational complexity, including problems that are computationally hard (even to approximate). Thus, a practically more relevant aspect is to empirically verify the formal statement, and test the capacity of MPNNs with RNI, in comparison to higher-order GNNs. Higher-order GNNs typically suffer from prohibitive space requirements, which is not the case for MPNNs with RNI, and this already makes them more viable in practice. As we discuss later in detail, our experiments demonstrate that MPNNs with RNI indeed combine expressiveness with efficiency in practice.

## 5 DATASETS FOR EVALUATING THE EXPRESSIVE POWER OF GNNS

GNN models are evaluated on prominent real-world datasets, such as IMDB, TU, and Proteins (Kersting et al., 2016). These datasets are not tailored for evaluating the expressive power of GNNs, as they do not contain instances or edge cases requiring expressiveness beyond 1-WL. In fact, higher-order models only marginally outperform MPNNs on these datasets (Maron et al., 2019a; Dwivedi et al., 2020), which further highlights their unsuitability for expressiveness evaluation.

We develop the datasets EXP and CEXP. EXP is designed to explicitly evaluate the expressiveness of GNN models, and consists of a set of graph instances $\{G_1 \ldots G_n, H_1 \ldots H_n\}$, such that each instance is a graph encoding of a propositional formula. The classification task is to determine whether the formula is satisfiable (SAT). Each pair $(G_i, H_i)$ respects the following properties: (i) $G_i$ and $H_i$ are non-isomorphic, (ii) $G_i$ and $H_i$ have different SAT outcomes, that is, $G_i$ encodes a satisfiable formula, while $H_i$ encodes an unsatisfiable formula, (iii) $G_i$ and $H_i$ are 1-WL indistinguishable, so are *guaranteed* to be classified in the same way by standard MPNNs, and (iv) $G_i$ and $H_i$ are 2-WL distinguishable, so *can* be classified differently by higher-order GNNs. Thanks to these properties, we can explicitly compare the performance of MPNNs with RNI to these higher-order models.

Ensuring these properties in EXP is highly non-trivial, and the construction of this dataset is cumbersome. Fundamentally, every $(G_i, H_i)$ is carefully constructed on top of a basic building block, the *core pair*, such that the 2 cores underlie $G_i$ and $H_i$, respectively. In this core pair, both cores are based on propositional clauses, such that one core is satisfiable and the other is not, and that these cores *exclusively* determine the satisfiability of $G_i$ (resp., $H_i$) and have graph encodings enabling all aforementioned properties. Core pairs, and their resulting graph instances in EXP are *planar* and are also carefully constrained to ensure they are 2-WL distinguishable. Hence, core pairs are key substructures within EXP, and distinguishing these cores is essential for good performance.

Building on EXP, CEXP includes instances with varying expressiveness requirements. Specifically, CEXP is a standard EXP dataset where 50% of all satisfiable graph pairs are modified, such that they become 1-WL distinguishable from their unsatisfiable counterparts, only differing from these by a small number of added edges. Hence, CEXP consists of 50% "corrupted" data, which can be distinguished and learned by a standard MPNN model (1-WL), which we label CORRUPT, and 50% unmodified data, generated analogously to EXP, and requiring expressive power beyond 1-WL, which we refer to as $\overline{\text{EXP}}$. Thus, CEXP contains the same core structures as EXP, but these now lead to different SAT values in $\overline{\text{EXP}}$ and CORRUPT, and this makes the overall learning task more challenging than learning $\overline{\text{EXP}}$ or CORRUPT in isolation. Complete details of the overall data generation process for both datasets can be found in the appendix.

## 6 EXPERIMENTAL EVALUATION

In this section, we first evaluate the practical effect of RNI on MPNN expressiveness based on EXP, and compare MPNN-RNI against established higher-order GNNs. We then extend our empirical analysis to CEXP. Both experiments are conducted using the following models:

- **1-WL GCN (1-GCN)**: A GCN with 8 distinct message passing iterations, ELU non-linearities (Clevert et al., 2016), 64-dimensional embeddings, and deterministic learnable initial node embeddings indicating node type. This model is guaranteed to achieve 50% accuracy on EXP.

- **GCN - Random node initialization (GCN-RNI)**: An analogous model to 1-GCN with an identical architecture, enhanced with RNI. We evaluate this model with four initialization distributions, namely the standard normal distribution $\mathcal{N}(0,1)$ (N), the uniform distribution over $[-1, 1]$ (U), Xavier normal (XN), and the Xavier uniform distribution (XU) (Glorot & Bengio, 2010). We denote the respective models GCN-RNI($D$), where $D \in \{\text{N,U,XN,XU}\}$.

- **GCN - Partial Random node initialization (GCN-$x$%RNI)**: A GCN-RNI model, where only a percentage $x$ of initial node embedding dimensions are randomized. That is, for $d$-dimensional node embeddings, GCN-$x$%RNI randomizes $\lfloor \frac{xd}{100} \rfloor$ dimensions, and sets the remaining dimensions deterministically from input features, namely, a one-hot representation of the two possible node types (literal and disjunction) in the input graph representation (see appendix for more details). We set $x$ to the extreme values 0 and 100%, 50%, as well as near-edge cases of 87.5% and 12.5%, respectively.

- **Provably Powerful Graph Network (PPGN)** (Maron et al., 2019a): A higher-order GNN with 2-WL expressive power, and which requires quadratic memory relative to the number of graph nodes. We set up PPGN with eight 400-dimensional computational blocks.

- **1-2-3-GCN-L**: A higher-order GNN (Morris et al., 2019) emulating 2-WL on 3-tuples of nodes. 1-2-3-GCN-L operates at increasingly coarse node granularities, starting with single nodes and rising to 3-tuples. The model first computes all possible 3-tuples of nodes, then represents them as standard graph nodes. These nodes are connected to one another following the 2-WL neighborhood definition, i.e., tuples that exchange messages in 2-WL are connected by an edge in 3-GCN. 3-GCN-L implements a *connected* relaxation of 2-WL, in that only 3-tuples forming a connected graph are used, which comes at the cost of some theoretical guarantees. Nonetheless, the computation and representation of all tuples still imposes a severe overhead relative to MPNNs. We set up 1-2-3-GCN-L with 64-dimensional embeddings, 3 message passing iterations at level 1, 2 at level 2 and 8 at level 3.

- **3-GCN**: A modification to 1-2-3-GCN-L, such that (i) only the 3rd level is used, and (ii) the *full* 2-WL procedure is implemented, i.e., all 3-tuples are computed, as in standard 2-WL, rather than only the connected ones.

### 6.1 EXPERIMENT 1: HOW DOES RNI IMPROVE MPNN EXPRESSIVENESS?

In this experiment, we evaluate GCNs using different RNI settings on EXP, and compare with standard GNNs and higher-order models. Specifically, we generate an EXP dataset consisting of 600 graph pairs, and discuss this generation in more detail in the appendix. Then, we evaluate all models on EXP using 10-fold cross-validation. We train 3-GCN for 100 epochs per fold, and all other systems for 500 epochs. *Mean test accuracy* across all validation folds is measured and reported.

Full test accuracy results for all models are reported in Table 1, and model convergence for 3-GCN and all GCN-RNI models are shown in Figure 2. In line with Theorem 4.1, GCN-RNI achieves a near-perfect performance on EXP, substantially surpassing 50%. Indeed, all fully randomized GCN-RNI models achieve a performance above 95% with all four RNI distributions. This finding supports observations made in related studies on RNI (Sato et al., 2020), which suggest that RNI enables (sub)structure detection beyond the theoretical limits of 1-WL. Empirically, we observed that GCN-RNI is highly sensitive to changes in learning rate, activation function, and/or randomization distribution, and required delicate tuning to achieve its best performance.

Table 1: Accuracy results on EXP.

| Model | Test Accuracy (%) |
|---|---|
| GCN-RNI(U) | 97.3 ± 2.55 |
| **GCN-RNI(N)** | **98.0 ± 1.85** |
| GCN-RNI(XU) | 97.0 ± 1.43 |
| GCN-RNI(XN) | 96.6 ± 2.20 |
| PPGN | 50.0 |
| 1-2-3-GCN-L | 50.0 |
| **3-GCN** | **99.7 ± 0.004** |

Surprisingly, PPGN does not achieve performance above 50%, despite being theoretically 2-WL expressive. Essentially, PPGN learns an approximation of 2-WL, based on power-sum multi-symmetric polynomials (PMP), but fails to distinguish EXP graph pairs, despite extensive training. This suggests that PPGNs struggle to learn the required PMPs, and we could not improve these results, both for training and testing, with hyperparameter tuning. Furthermore, as mentioned in Section 3, PPGN requires exponentially many data samples in the size of the input graph (Puny et al., 2020) for learning. Hence, PPGN is likely struggling to discern between EXP graph pairs due to the smaller sample size and variability of the dataset. 1-2-3-GCN-L also only achieves 50% accuracy, which can be attributed to theoretical model limitations. Indeed, the local and connected algorithm drops necessary information to distinguish graph pairs, as it only considers 3-tuples of nodes that form a connected sub-graph. Thus, 1-2-3-GCN-L discards disconnected 3-tuples that crucially are where the difference between the EXP cores lies. This further highlights the difficulty of EXP instances, as even a relaxation of 2-WL costs the model the ability to achieve above-random performance. Note that 3-GCN achieves near-perfect performance, as it explicitly has the sufficient theoretical power needed for the task, irrespective of learning constraints, and must only learn appropriate injective aggregation functions for neighbor aggregation (Xu et al., 2019).

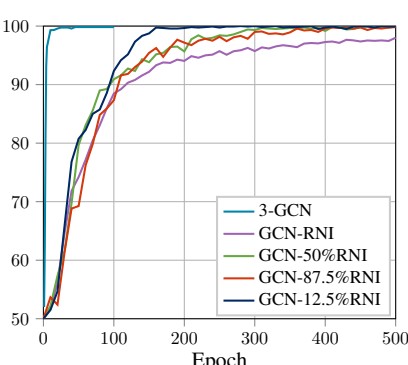

Figure 2: Learning curves on EXP.

In terms of model convergence, we observe that 3-GCN converges significantly faster than all GCN-RNI models, for all randomization percentages. Indeed, 3-GCN only requires about 10 epochs to achieve its optimal performance, whereas GCN-RNI models all require in excess of 100 epochs. The rp Intuitively, the slower convergence of GCN-RNI can be attributed to a significantly harder learning task compared to 3-GCN: Whereas 3-GCN must learn from a deterministic set of node embeddings, and is naturally capable of discerning between dataset cores, GCN-RNI relies on RNI to discern between data points in EXP, via an artificial node ordering. This in turn implies that GCN-RNI must first leverage RNI to detect structure, then subsequently learn robustness against the variability of RNI, which makes the learning task for GCN-RNI especially challenging.

Our findings suggest that RNI can practically improve the expressiveness of MPNNs, and make them competitive with higher-order models, despite being significantly less demanding computationally. Indeed, for a typical EXP instance with 50 nodes, GCN-RNI only requires 3200 parameters (using

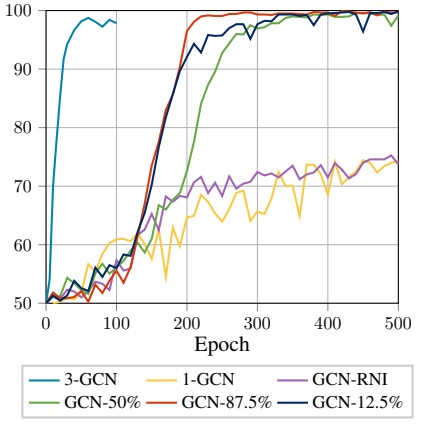 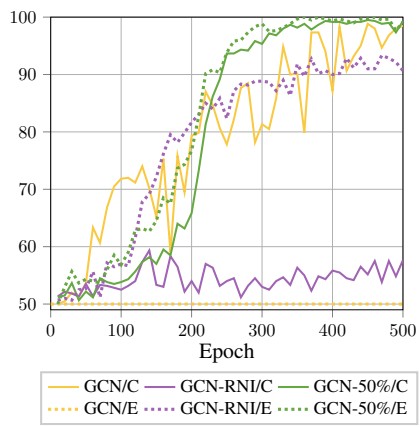

(a) Learning curves for all GCN-RNI models and 3-GCN on CExp.

(b) Learning curves for CExp, split across $\overline{\text{Exp}}$ (/E) and Corrupt (/C).

Figure 3: Model convergence results for Experiment 2 on CExp on all models.

64-dimensional embeddings), whereas 3-GCN requires 1,254,400 parameters. Nonetheless, GCN-RNI performs comparably to 3-GCN, and, unlike the latter model, can easily scale to larger instances exceeding the range used in our datasets. This increase in expressive power, however, comes at the cost of slower convergence. Even so, RNI proves to be a promising direction for building scalable yet powerful MPNNs.

## 6.2 Experiment 2: How does RNI affect MPNN on more variable datasets?

In Experiment 1, we observed that RNI practically improves the expressive power of GCNs over Exp. However, Exp is solely designed for expressiveness evaluation, and this leaves multiple questions open: How does RNI impact learning when data contains instances with varying expressiveness requirements, and how does RNI affect model generalization on more variable datasets? We experiment with CExp to explicitly address these questions.

Analogously to Experiment 1, we generate an Exp dataset with 600 pairs of graphs. Then, we create CExp by selecting 300 graph pairs and modifying their satisfiable graph, yielding Corrupt. CExp is well-suited for evaluating the efficacy of RNI more holistically, as it allows (i) the evaluation of the contribution of RNI on $\overline{\text{Exp}}$ conjointly with a second learning task on Corrupt involving very similar core structures, and (ii) a study of the effect of different degrees of randomization on overall and subset-specific model performance.

In this experiment, we train GCN-RNI (with varying randomization degrees) and 3-GCN on CExp, and compare their test accuracy across all cross-validation splits. For GCN-RNI models, we observe the effect of RNI on specifically learning $\overline{\text{Exp}}$ and Corrupt, and the interplay between these two tasks. In all experiments, we exclusively use normal distribution initialization, given its strong performance in Experiment 1.

The learning curves of all GCN-RNI and 3-GCN on CExp are shown in Figure 3a, and the same curves for the $\overline{\text{Exp}}$ and Corrupt subsets are shown in Figure 3b. As on Exp, we observe that 3-GCN converges very quickly, exceeding 90% test accuracy within 25 epochs on CExp. By contrast, GCN-RNI, for all randomization levels, converges much slower, around after 200 epochs, despite the small size of input graphs (~70 nodes at most). Furthermore, fully randomized GCN-RNI performs worse than partly randomized GCN-RNI models, particularly on CExp, due to its weak performance on Corrupt, as shown in Figure 3b.

First, we observe that partial randomization can significantly improve model performance. This can clearly be seen on CExp, in Figure 3a and Figure 3b, where GCN-12.5%RNI and GCN-87.5%RNI achieve the best performance, by far outperforming GCN-RNI, which struggles on Corrupt. This can be attributed to having a better inductive bias than a fully randomized model. Indeed, GCN-12.5%RNI has mostly deterministic node embeddings, which simplifies learning over Corrupt.

This also applies to GCN-87.5%RNI, where the number of deterministic dimensions, though small, remains sufficient for learning over CORRUPT. Both models also benefit from randomization to perform strongly on $\overline{\text{EXP}}$, and have sufficient randomization to perform similarly to a fully randomized GCN. GCN-12.5%RNI and GCN-87.5%RNI effectively achieve the best of both worlds on CEXP, leveraging inductive bias from deterministic node embeddings, while harnessing the power of random embeddings to perform strongly on $\overline{\text{EXP}}$. This is best shown in Figure 3b, where standard GCN fails to learn $\overline{\text{EXP}}$, fully randomized GCN-RNI struggles to learn CORRUPT, and the semi-randomized GCN-50%RNI achieves perfect performance on both subsets. Overall, this is a surprising finding, as it suggests that MPNNs can perform significantly better with partial, and even small, amounts of randomization.

Second, we observe that the fully randomized GCN-RNI performs substantially worse than its partially randomized counterparts. Whereas fully randomized GCN-RNI only performs marginally worse on EXP (cf. Figure 2) than partially randomized models, this gap is very large on CEXP, primarily due to CORRUPT. This observation concurs with the earlier idea of inductive bias: Fully randomized GCN-RNI loses all node type information, which is valuable for making robust and consistent decisions, and therefore struggles to match 3-GCN and partially randomized models. Indeed, the model fails to achieve even 60% accuracy on CORRUPT, where other models are near perfect, and also relatively struggles on $\overline{\text{EXP}}$, only reaching 91% accuracy and converging slower.

Third, all GCN-RNI models, at all randomization levels, converge significantly slower on both datasets than 3-GCN, similarly to Experiment 1. However, an interesting phenomenon can be seen on CEXP: All GCN-RNI models hover around 55% accuracy within the first 100 epochs over CEXP (cf. Figure 3a), suggesting a struggle jointly fitting both CORRUPT and $\overline{\text{EXP}}$, before these models ultimately improve. This, however, is not observed with 3-GCN. Unlike on EXP, randomness is not necessarily beneficial on CEXP, as it can hurt performance on CORRUPT. Hence, RNI-enhanced models must additionally learn to isolate deterministic dimensions for CORRUPT, and randomized dimensions for $\overline{\text{EXP}}$. These findings consolidate the earlier observations made on EXP on the impact of RNI on MPNN learning behavior, and highlight that the variability and slower learning for RNI also hinges on the variability and complexity of the input dataset.

Finally, we observe that both fully randomized GCN-RNI, and, surprisingly, 1-GCN, struggle to learn CORRUPT relative to partially randomized GCN-RNI. We can also observe that 1-GCN does not present a "struggle" phase, and begins improving consistently from the start of training. These observations can be attributed to key conceptual , but very distinct hindrances impeding both models. In the case of 1-GCN, the model is jointly trying to learn both $\overline{\text{EXP}}$ and CORRUPT, when it is proven that it cannot fit the former. This joint optimization severely hinders CORRUPT learning, as data pairs from both subsets are highly similar, and share identically generated UNSAT graphs (cf. Appendix). Hence, 1-GCN, in attempting to fit SAT graphs from both subsets, knowing it cannot distinguish $\overline{\text{EXP}}$ pairs, struggles to learn the simpler difference in CORRUPT pairs. For GCN-RNI, the model discards key type information, so must only rely on structural differences to learn CORRUPT, which impedes its convergence. All in all, this further consolidates the promise of partial RNI as a means to combine the strengths of both deterministic and random features.

Further to the earlier two experiments, we also conducted analogous experiments using sparser analogs of the datasets EXP and CEXP. In these cases, we observed similar behavior, albeit with slower convergence overall. More details on these experiments can be found in the appendix.

## 7 SUMMARY AND OUTLOOK

We studied the expressive power of MPNNs with RNI, and showed that these are universal models. We empirically evaluated this model on carefully designed datasets, and observed that RNI practically improves the learning abilities of MPNNs for challenging data, though it does slow down model convergence owing to the need to learn robustness against random variability. Our work delivers a strong theoretical result, supported by empirical evaluation and practical insights, to rigorously quantify the effect of RNI on GNNs. Somewhat surprisingly, our experiments suggest that partial randomization may be the best strategy in most practical scenarios. An important direction for future work is to theoretically study the sensitivity of RNI to model architectures and initialization distributions, to yield a more complete understanding of the benefits and limitations of RNI.

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

# A    APPENDIX

## A.1    PROPOSITIONAL LOGIC

We briefly present propositional logic, which underpins the dataset generation. Let $S$ be a (finite) set $S$ of propositional variables. A *literal* is defined as $v$, or $\bar{v}$ (resp., $\neg v$), where $v \in S$. A disjunction of literals is a *clause*. The *width* of a clause is defined as the number of literals it contains. A formula $\varphi$ is in *conjunctive normal form (CNF)* if it is a conjunction of clauses. A CNF has width $k$ if it contains clauses of width at most $k$, and is referred to as a $k$–CNF. To illustrate, the formula $\varphi = (x_1 \vee \neg x_3) \wedge (x_4 \vee x_1)$ is a CNF with clauses of width 2.

An assignment $\nu : S \mapsto \{0, 1\}$ maps variables to False (0), or True (1), and *satisfies* $\varphi$, which we denote by $\nu \vDash \varphi$, in the usual sense, where $\vDash$ is propositional entailment. Given a propositional formula $\varphi$, the *satisfiability problem*, commonly known as SAT, consists of determining whether $\varphi$ admits a satisfying assignment, and is NP-complete (Cook, 1971).

## A.2    PROOF OF THEOREM 4.1

We first prove a Boolean version of the theorem.

**Lemma A.1.** *Let $n \geq 1$, and let $f : \mathcal{G}_n \to \{0, 1\}$ be an invariant Boolean function. Then, for all $\epsilon, \delta > 0$ there is a MPNN with RNI that $(\epsilon, \delta)$-approximates $f$.*

To prove this lemma, we use a logical characterization of the expressiveness of MPNNs, which we always assume to admit global readouts. Let $\mathsf{C}$ be the extension of first-order predicate logic using counting quantifiers of the form $\exists^{\geq k} x$ for $k \geq 0$, where $\exists^{\geq k} x \varphi(x)$ means that there are at least $k$ elements $x$ satisfying $\varphi$.

For example, consider the formula

$$\varphi(x) := \neg \exists^{\geq 3} y \big( E(x, y) \wedge \exists^{\geq 5} z E(y, z) \big). \tag{A.1}$$

This is a formula in the language of graphs; $E(x, y)$ means that there is an edge between the nodes interpreting $x$ and $y$. For a graph $G$ and a vertex $v \in V(G)$, we have $G \vDash \varphi(v)$ ("$G$ satisfies $\varphi$ if the variable $x$ is interpreted by the vertex $v$") if and only if $v$ has at most 2 neighbors in $G$ that have degree at least 5.

We will not only consider formulas in the language of graphs, but also formulas in the *language of colored graphs*, where in addition to the binary edge relation we also have unary relations, that is, sets of nodes, which we may view as colors of the nodes. For example, the formula

$$\psi(x) := \exists^{\geq 4} y \big( E(x, y) \wedge RED(y) \big)$$

says that node $x$ has at least 4 red neighbors (more precisely, neighbors in the unary relation *RED*). Formally, we assume we have fixed infinite list $R_1, R_2, \ldots$ of color symbols that we may use in our formulas. Then a *colored graph* is a graph together with a mapping that assigns a finite set $\rho(v)$ of colors $R_i$ to each vertex (so we allow one vertex to have more than one, but only finitely many, colors).

A *sentence* (of the logic $\mathsf{C}$ or any other logic) is a formula without free variable. Thus a sentence expresses a property of a graph, which we can also view as a Boolean function. For a sentence $\varphi$ we denote this function by $\llbracket \varphi \rrbracket$. If $\varphi$ is a sentence in the language of (colored) graphs, then for every (colored) graph $G$ we have $\llbracket \varphi \rrbracket(G) = 1$ if $G \vDash \varphi$ and $\llbracket \varphi \rrbracket(G) = 0$ otherwise.

It is easy to see that $\mathsf{C}$ is only a syntactic extension of first order logic $\mathsf{FO}$—for every $\mathsf{C}$-formula there is a logically equivalent $\mathsf{FO}$-formula. To see this, note that we can simulate $\exists^{\geq k} x$ by $k$ ordinary existential quantifiers: $\exists^{\geq k} x$ is equivalent to $\exists x_1 \ldots \exists x_k \big( \bigwedge_{1 \leq i < j \leq k} x_i \neq x_j \wedge \bigwedge_{1 \leq i \leq k} \varphi(x_i) \big)$. However, counting quantifiers add expressiveness if we restrict the number of variables. The $\mathsf{C}$-formula $\exists^{\geq k} x \; (x = x)$ (saying that there are at least $k$ vertices) with just one variable is not equivalent to any $\mathsf{FO}$-formula using less than $k$ variables. By $\mathsf{C}^k$ we denote the fragment of $\mathsf{C}$ consisting of all formulas with at most $k$ variables.

For example, the formula $\varphi(x)$ in (A.1) is in $\mathsf{C}^3$, but not in $\mathsf{C}^2$. But $\varphi(x)$ is equivalent to the following formula $\varphi'(x)$ in $\mathsf{C}^2$:

$$\varphi'(x) := \neg \exists^{\geq 3} y \big( E(x,y) \wedge \exists^{\geq 5} x E(y,x) \big). \tag{A.2}$$

The fragments $\mathsf{C}^k$ are interesting for us, because their expressiveness corresponds to that of $(k-1)$-WL and hence to that of $k$-GNNs. More precisely, for all $k \geq 2$, two graphs $G$ and $H$ satisfy the same $\mathsf{C}^k$-sentences if and only if $(k-1)$-WL does not distinguish them (Cai et al., 1992). By the results of (Morris et al., 2019; Xu et al., 2019) this implies, in particular, that two graphs are indistinguishable by all MPNNs if and only if they satisfy the same $\mathsf{C}^2$-sentences. Barceló et al. (2020) strengthened this result and showed that every $\mathsf{C}^2$-sentence can be simulated by an MPNN.

**Lemma A.2** (Barceló et al. 2020). *For every $\mathsf{C}^2$-sentence $\varphi$ and every $\epsilon > 0$ there is an MPNN that $\epsilon$-approximates $[\![\varphi]\!]$.*

Since here we are talking about deterministic MPNNs, there is no randomness involved, and we just say *"$\epsilon$-approximates"* instead of *"$(\epsilon, 1)$-approximates"*.

Lemma A.2 not only holds for sentences in the language of graphs, but also for sentences in the language of colored graphs. Let us briefly discuss the way MPNNs access such colors. We encode the colors using one-hot vectors that are part of the initial states of the nodes. For example, if we have a formula that uses color symbols among $R_1, \ldots, R_k$, then we reserve $k$ places in the initial state $\boldsymbol{x}_v = (x_{v1}, \ldots, x_{v\ell})$ of each vertex $v$ (say, for convenience, $x_{v1}, \ldots, x_{vk}$) and we initialize $\boldsymbol{x}_v$ by letting $x_{vi} = 1$ if $v$ is in $R_i$ and $x_{vi} = 0$ otherwise.

Let us call a colored graph $G$ *individualized* if for any two distinct vertices $v, w \in V(G)$ the sets $\rho(v), \rho(w)$ of colors they have are distinct. Let us say that a sentence $\chi$ *identifies* a (colored) graph $G$ if for all (colored) graphs $H$ we have $H \vDash \chi$ if and only if $H$ is isomorphic to $G$.

**Lemma A.3.** *For every individualized colored graph $G$ there is a $\mathsf{C}^2$-sentence $\chi_G$ that identifies $G$.*

*Proof.* Let $G$ be an individualized graph. For every vertex $v \in V(G)$, let

$$\alpha_v(x) := \bigwedge_{R \in \rho(v)} R(x) \wedge \bigwedge_{R \in \{R_1, \ldots, R_k\} \setminus \rho(x)} \neg R(x).$$

Then $v$ is the unique vertex of $G$ such that $G \vDash \alpha_v(v)$. For every pair $v, w \in V(G)$ of vertices, we let

$$\beta_{vw}(x,y) := \begin{cases} \alpha_v(x) \wedge \alpha_w(y) \wedge E(x,y) & \text{if } (v,w) \in E(G), \\ \alpha_v(x) \wedge \alpha_w(y) \wedge \neg E(x,y) & \text{if } (v,w) \notin E(G). \end{cases}$$

We let

$$\chi_G := \bigwedge_{v \in V(G)} \big( \exists x \alpha_v(x) \wedge \neg \exists^{\geq 2} x \alpha_v(x) \big) \wedge \bigwedge_{v, w \in V(G)} \exists x \exists y \beta_{vw}(x,y).$$

It is easy to see that $\chi_G$ identifies $G$. $\qquad\square$

For $n, k \in \mathbb{N}$, we let $\mathcal{G}_{n,k}$ be the class of all individualized colored graphs that only use colors among $R_1, \ldots, R_k$.

**Lemma A.4.** *Let $h : \mathcal{G}_{n,k} \to \{0,1\}$ be an invariant Boolean function. Then there exists a $\mathsf{C}^2$-sentence $\psi_h$ such that for all $G \in \mathcal{G}_{n,k}$ it holds that $[\![\psi_h]\!](G) = h(G)$.*

*Proof.* Let $\mathcal{H} \subseteq \mathcal{G}_{n,k}$ be the subset consisting of all graphs $H$ with $h(H) = 1$. We let

$$\psi_h := \bigvee_{H \in \mathcal{H}} \chi_H.$$

We eliminate duplicates in the disjunction. Since up to isomorphism, the class $\mathcal{G}_{n,k}$ is finite, this makes the disjunction finite and hence $\psi_h$ well-defined. $\qquad\square$

The *restriction* of a colored graph $G$ is the underlying plain graph, that is, the graph $G^\vee$ obtained from the colored graph $G$ by forgetting all the colors. Conversely, a colored graph $G^\wedge$ is an *expansion* of a plain graph $G$ if $G = (G^\wedge)^\vee$.

**Corollary A.1.** *Let $f : \mathcal{G}_n \to \{0,1\}$ be an invariant Boolean function. Then there exists a $\mathsf{C}^2$-sentence $\varphi_f^\wedge$ (in the language of colored graphs) such that for all $G \in \mathcal{G}_{n,k}$ it holds that $[\![\psi_f^\wedge]\!](G) = f(G^\vee)$.*

Towards proving Lemma A.1, we fix an $n \geq 1$ and a $\epsilon, \delta > 0$. We let

$$c := \left\lceil \frac{2}{\delta} \right\rceil \quad \text{and} \quad k := c^2 \cdot n^3$$

The technical details of the proof of Lemma A.1 and Theorem 4.1 depend on the exact choice of the random initialization and the activation functions used in the neural networks, but the idea is always the same. For simplicity, we assume that we initialize the states $\boldsymbol{x}_v = (x_{v1}, \ldots, x_{v\ell})$ of all vertices to $(r_v, 0, \ldots, 0)$, where $r_v$ for $v \in V(G)$ are chosen independently uniformly at random from $[0,1]$. As our activation function $\sigma$, we choose the linearized sigmoid function defined by $\sigma(x) = 0$ for $x < 0$, $\sigma(x) = x$ for $0 \leq x < 1$, and $\sigma(x) = 1$ for $x \geq 1$.

**Lemma A.5.** *Let $r_1, \ldots, r_n$ be chosen independently uniformly at random from the interval $[0,1]$. For $1 \leq i \leq n$ and $1 \leq j \leq c \cdot n^2$, let*

$$s_{ij} := k \cdot r_i - (j-1) \cdot \frac{k}{c \cdot n^2}.$$

*Then with probability greater than $1 - \delta$, the following conditions are satisfied.*

*(i) For all $i \in \{1, \ldots, n\}, j \in \{1, \ldots, c \cdot n^2\}$ it holds that $\sigma(s_{ij}) \in \{0,1\}$.*

*(ii) For all distinct $i, i' \in \{1, \ldots, n\}$ there exists a $j \in \{1, \ldots, c \cdot n^2\}$ such that $\sigma(s_{ij}) \neq \sigma(s_{i'j})$.*

*Proof.* For every $i$, let $p_i := \lfloor r_i \cdot k \rfloor$. Since $k \cdot r_i$ is uniformly random from the interval $[0,k]$, the integer $p_i$ is uniformly random from $\{0, \ldots, k-1\}$. Observe that $0 < \sigma(s_{ij}) < 1$ only if $p_i - (j-1) \cdot \frac{k}{c \cdot n^2} = 0$ (here we use the fact that $k$ is divisible by $c \cdot n^2$). The probability that this happens is $\frac{1}{k}$. Thus, by the Union Bound,

$$\Pr\left(\exists i, j : 0 < \sigma(s_{ij}) < 1\right) \leq \frac{c \cdot n^3}{k}. \tag{A.3}$$

Now let $i, i'$ be distinct and suppose that $\sigma(s_{ij}) = \sigma(s_{i'j})$ for all $j$. Then for all $j$ we have $s_{ij} \leq 0 \iff s_{i'j} \leq 0$ and therefore $\lfloor s_{ij} \rfloor \leq 0 \iff \lfloor s_{i'j} \rfloor \leq 0$. This implies

$$\forall j \in \{1, \ldots, c \cdot n^2\} : \quad p_i \leq (j-1) \cdot \frac{k}{c \cdot n^2} \iff p_{i'} \leq (j-1) \cdot \frac{k}{c \cdot n^2}. \tag{A.4}$$

Let $j^* \in \{1, \ldots, c \cdot n^2\}$ such that $p_i \in \left\{(j^*-1) \cdot \frac{k}{c \cdot n^2}, \ldots, j^* \cdot \frac{k}{c \cdot n^2} - 1\right\}$. Then by (A.4) we have $p'_i \in \left\{(j^*-1) \cdot \frac{k}{c \cdot n^2}, \ldots, j^* \cdot \frac{k}{c \cdot n^2} - 1\right\}$. As $p_{i'}$ is independent of $p_i$ and hence of $j^*$, the probability that this happens is at most $\frac{1}{k} \cdot \frac{k}{c \cdot n^2} = \frac{1}{c \cdot n^2}$. This proves that for all distinct $i, i'$ the probability that $\sigma(s_{ij}) = \sigma(s_{i'j})$ is at most $\frac{1}{c \cdot n^2}$. Hence, again by the Union Bound,

$$\Pr\left(\exists i \neq i' \forall j : \sigma(s_{ij}) = \sigma(s_{i'j})\right) \leq \frac{1}{c}. \tag{A.5}$$

(A.3) and (A.5) imply that the probability that either (i) or (ii) is violated is at most

$$\frac{c \cdot n^3}{k} + \frac{1}{c} \leq \frac{2}{c} \leq \delta. \qquad \square$$

*Proof of Lemma A.1.* For given function $f : \mathcal{G}_n \to \{0,1\}$, we choose the sentence $\psi_f^\wedge$ according to Corollary A.1. Applying Lemma A.2 to this sentence and $\epsilon$, we obtain an MPNN $\mathcal{N}_f$ that on a colored graph $G \in \mathcal{G}_{n,k}$ computes an $\epsilon$-approximation of $f(G^\vee)$.

Without loss of generality, we assume that the vertex set of the input graph to our MPNN is $\{1, \ldots, n\}$. We choose $\ell$ (the dimension of the state vectors) in such a way that $\ell \geq c \cdot n^2$ and $\ell$ is at least as large as the dimension of the state vectors of $\mathcal{N}_f$. Recall that the state vectors are

initialized as $\boldsymbol{x}_i^{(0)} = (r_i, 0, \ldots, 0)$ for values $r_i$ chosen independently uniformly at random from the interval $[0, 1]$.

In the first step, our MPNN computes the purely local transformation (no messages need to be passed) that maps $\boldsymbol{x}_i^{(0)}$ to $\boldsymbol{x}_i^{(1)} = (x_{i1}^{(1)}, \ldots, x_{i\ell}^{(1)})$ with

$$x_{ij}^{(1)} = \begin{cases} \sigma\left(k \cdot r_i - (j-1) \cdot \frac{k}{c \cdot n^2}\right) & \text{for } 1 \le j \le c \cdot n^2, \\ 0 & \text{for } c \cdot n^2 + 1 \le j \le \ell. \end{cases}$$

Since we treat $k, c, n$ as constants, the mapping $r_i \mapsto k \cdot r_i - (j-1) \cdot \frac{k}{c \cdot n^2}$ is just a linear mapping applied to $r_i = x_{i1}^{(0)}$.

By Lemma A.5, with probability at least $1 - \delta$, the vectors $\boldsymbol{x}_i^{(1)}$ are mutually distinct $\{0, 1\}$-vectors, which we view as encoding a coloring of the input graph with colors from $R_1, \ldots, R_k$. Let $G^\wedge$ be the resulting colored graph. Since the vectors $\boldsymbol{x}_i^{(0)}$ are mutually distinct, $G^\wedge$ is individualized and thus in the class $\mathcal{G}_{n,k}$. We now apply the MPNN $\mathcal{N}_f$, and it computes a value $\epsilon$-close to $[\![\psi_f^\wedge]\!](G^\wedge) = f((G^\wedge)^\vee) = f(G)$. $\qquad\square$

*Proof of Theorem 4.1.* Let $f : \mathcal{G}_n \to \mathbb{R}$ be invariant. Since $\mathcal{G}_n$ is finite, the range $Y := f(\mathcal{G}_n)$ is finite. To be precise, we have $N := |Y| \le |\mathcal{G}_n| = 2^{\binom{n}{2}}$.

Say, $Y = \{y_1, \ldots, y_N\}$. For $i = 1, \ldots, N$, let $g_i : \mathcal{G}_n \to \{0, 1\}$ be the Boolean function defined by

$$g_i(G) = \begin{cases} 1 & \text{if } f(G) = y_i, \\ 0 & \text{otherwise.} \end{cases}$$

Note that $g_i$ is invariant.

Let $\epsilon, \delta > 0$ and $\epsilon' := \frac{\epsilon}{\max Y}$ and $\delta' := \frac{\delta}{N}$. By Lemma A.1, for every $i \in \{1, \ldots, N\}$ there is an MPNN with RNI $\mathcal{N}_i$ that $(\epsilon', \delta)$-approximates $g_i$. Putting all the $\mathcal{N}_i$ together, we obtain an invariant MPNN $\mathcal{N}$ that computes a function $g : \mathcal{G}_n \to \{0, 1\}^N$. We only need to apply the linear transformation

$$\boldsymbol{x} \mapsto \sum_{i=1}^N x_i \cdot y_i$$

to the output of $\mathcal{N}$ to obtain the desired approximation of $f$. $\qquad\square$

**Remark 1.** Obviously, our construction yields MPNNs with a prohibitively large state space. In particular, this is the case for the brute force step from Boolean to general functions. We doubt that there are much more efficient approximators, after all we make no assumption whatsoever on the function $f$.

The approximation of Boolean functions is more interesting. It may still happen that the GNNs get exponentially large in $n$; this seems unavoidable. However, the nice thing here is that our construction is very adaptive and tightly linked to the descriptive complexity of the function we want to approximate. This deserves a more thorough investigation, which we leave for future work.

As opposed to other universality results for GNNs, our construction needs no higher-order tensors defined on tuples of nodes, with practically infeasible space requirements on all but very small graphs. Instead, the complexity of our construction goes entirely into the dimension of the state space. The advantage of this is that we can treat this dimension as a hyperparameter that we can easily adapt and that gives us more fine-grained control over the space requirements. Our experiments show that usually in practice a small dimension already yields very powerful networks.

**Remark 2.** In our experiments, we found that a partial random initialization, which only assigns random values to a fraction of all node embedding vectors, often yields very good results, sometimes better than a full random initialization. There is plausibility to this from a theoretical perspective. For most graphs, we do not lose much by only initializing a small fraction of vertex embeddings, because in a few message-passing rounds GNNs can propagate the randomness and, referring our construction above, individualize the full input graph. On the other hand, we reduce the amount of noise our models have to handle when we only randomize partially.

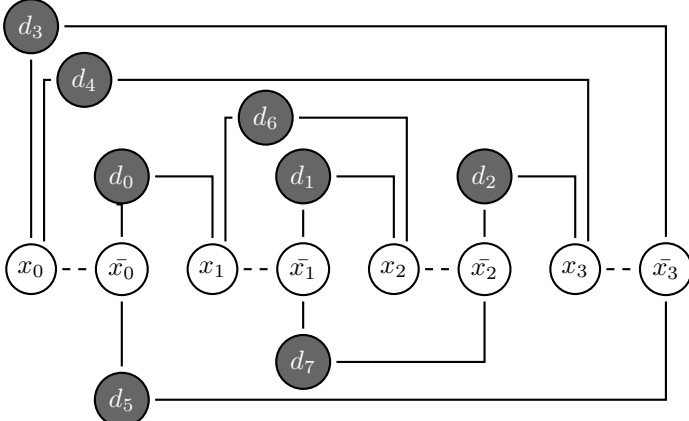

(a) The encoding of the formula $\varphi_1$.

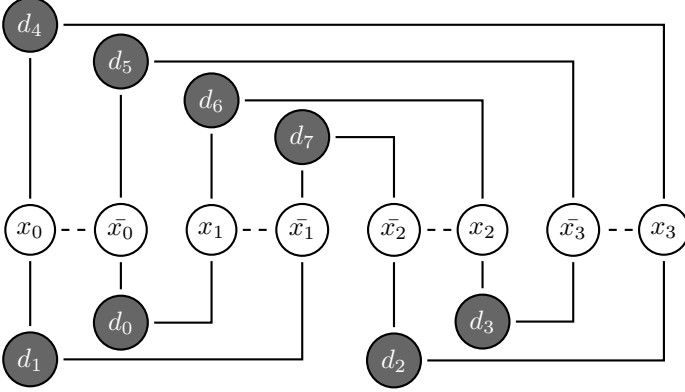

(b) The encoding of the formula $\varphi_2$.

Figure 4: Illustration of planar embeddings for the formulas $\varphi_1$ and $\varphi_2$ for $n = 2$.

## A.3 DETAILS OF DATASET CONSTRUCTION

There is an interesting universality result for functions defined on planar graphs. It is known that 3-WL can distinguish between any pair of planar graphs (Kiefer et al., 2019). Since 4-GCNs can simulate 3-WL, this implies that functions defined on planar graphs can be approximated by 4-GCNs. This result can be extended to much wider graph classes, including all graph classes excluding a fixed graph as a minor (Grohe, 2017).

Inspired by this, we generate planar instances, and ensure that they can be distinguished by 2-WL, by carefully constraining these instances further. Hence, any GNN with 2-WL expressive power can approximate solutions to these planar instances. This, however, does not imply that these GNNs will solve EXP in practice, but only that an appropriate approximation function exists and can theoretically be learned.

### A.3.1 CONSTRUCTION OF EXP

We now explain the construction and composition of EXP. Fundamentally, EXP consists of two main components, (i) a pair of *cores*, which are non-isomorphic, planar, 1-WL indistinguishable, 2-WL distinguishable, and decide the satisfiability of every instance, and (ii) an additional randomly generated and satisfiable *planar component*, identically added to the core pair, to add variability to EXP and make learning more challenging. We first present both components, and then provide further details about graph encoding and planar embeddings.

**Core pair.** In EXP, a core pair consists of two CNF formulas $\varphi_1, \varphi_2$, both defined using $2n$ variables, $n \in \mathbb{N}^+$, such that $\varphi_1$ is unsatisfiable and $\varphi_2$ is satisfiable, and such that their graph encodings are 1-WL indistinguishable and planar. $\varphi_1$ and $\varphi_2$ are constructed using two structures which we refer to as *variable chains* and *variable bridges* respectively.

A *variable chain* $\varphi_{chain}$ is defined over a set of $n \geq 2$ Boolean variables, and imposes that all variables be equally set. The variable chain can be defined in increasing or decreasing order over these variables. More specifically, given variables $x_i, ..., x_j$,

$$\text{Chain}_{\text{Inc}}(i, j) = \bigwedge_{k=i}^{j-1} (\bar{x_k} \vee x_{i+(k+1)\%(j-i+1)}), \text{ and} \tag{A.6}$$

$$\text{Chain}_{\text{Dec}}(i, j) = \bigwedge_{k=i}^{j-1} (x_k \vee \bar{x}_{i+(k+1)\%(j-i+1)}). \tag{A.7}$$

Additionally, a *variable bridge* is defined over an even number of variables $x_0, ..., x_{2n-1}$, as

$$\varphi_{bridge} = \bigwedge_{i=0}^{n-1} \big( (x_i \vee x_{2n-1-i}) \wedge (\bar{x}_i \vee \bar{x}_{2n-1-i}) \big). \tag{A.8}$$

A variable bridge makes the variables it connects forcibly have opposite values, e.g., $x_0 = \bar{x_1}$ for $n = 1$. We denote a variable bridge over $x_0, ..., x_{2n-1}$ as $\text{Bridge}(2n)$.

To get $\varphi_1$ and $\varphi_2$, we define $\varphi_1$ as a variable chain and bridge on all variables, yielding contrasting and unsatisfiable constraints. To define $\varphi_2$, we "cut" the chain in half, such that the first $n$ variables can differ from the latter $n$, satisfying the bridge. The second half of the "cut" chain is then flipped to a decrementing order, which preserves the satisfiability of $\varphi_2$, but maintains the planarity of the resulting graph. More specifically, this yields:

$$\varphi_1 = \text{Chain}_{\text{Inc}}(0, 2n) \wedge \text{Bridge}(2n), \text{ and} \tag{A.9}$$
$$\varphi_2 = \text{Chain}_{\text{Inc}}(0, n) \wedge \text{Chain}_{\text{Dec}}(n, 2n) \wedge \text{Bridge}(2n). \tag{A.10}$$

**Planar component.** Following the generation of $\varphi_1$ and $\varphi_2$, a disjoint satisfiable planar graph component $\varphi_{\text{planar}}$ is added. $\varphi_{\text{planar}}$ shares no variables or disjunctions with the cores, so is primarily introduced to create noise and make learning more challenging. $\varphi_{\text{planar}}$ is generated starting from random 2-connected (i.e., at least 2 edges must be removed to disconnect a component within the graph) bipartite planar graphs from the Plantri tool (Brinkmann et al., 2007), such that (i) the larger set of nodes in the graph is the variable set[3], (ii) highly-connected disjunctions are split in a planarity-preserving fashion to maintain disjunction widths not exceeding 5, (iii) literal signs for variables are uniformly randomly assigned, and (iv) redundant disjunctions, if any, are removed. If this $\varphi_{\text{planar}}$ is satisfiable, then it is accepted and used. Otherwise, the formula is discarded and a new $\varphi_{\text{planar}}$ is analogously generated until a satisfiable formula is produced.

Since the core pair and $\varphi_{\text{planar}}$ are disjoint, it is easy to deduce that the graph encoding of $\varphi_{\text{planar}} \wedge \varphi_1$ and $\varphi_{\text{planar}} \wedge \varphi_2$ are both planar and 1-WL indistinguishable. Furthermore, $\varphi_{\text{planar}} \wedge \varphi_1$ is satisfiable, and $\varphi_{\text{planar}} \wedge \varphi_2$ is not. Hence, the introduction of $\varphi_{\text{planar}}$ maintains all the desirable core properties, all while making any generated EXP dataset more challenging.

The structural properties of the cores, combined with the combinatorial difficulty of SAT, make EXP a challenging dataset. For example, even minor formula changes, such as flipping a literal, can lead to a change in the SAT outcome, which enables the creation of near-identical, yet semantically different instances. Moreover, SAT is NP-complete (Cook, 1971), and remains so on planar instances (Hunt III et al., 1998). Hence, EXP is cast to be challenging, both from an expressiveness and computational perspective.

**Remark 3.** Intuitively, $\varphi_1$ and $\varphi_2$, generated as described, can be distinguished by 2-WL, as 2-WL can detect the break in cycles resulting from the aforementioned "cut". In other words, 2-WL can identify that the chain has been broken in between these two formulas, and thus will return distinct colourings. Hence, $\varphi_1$ and $\varphi_2$ can be distinguished by 3-GCNs.

---

[3]Ties are broken arbitrarily if the two sets are equally sized.

**Graph encoding.** We use the following graph encoding, denoted by $Enc$: (i) Every variable is encoded by two nodes, representing its positive and negative literals, and connected by an edge, (ii) Every disjunction is represented by a node, and an edge connects a literal node to a disjunction node if the literal appears in the disjunction, and (iii) Variable and disjunction nodes are encoded with different types. We opt for this encoding, as it is commonly used in the literature (Selsam et al., 2019), and is sufficient, for the sake of our empirical evaluation, to yield planar encodings for EXP graph pairs.

**Planar embeddings for core pair.** We show planar embeddings for $Enc(\varphi_1)$ and $Enc(\varphi_2)$ for $n = 2$ in Figure 4, and these embeddings can naturally be extended to any $n$. $Enc(\varphi_1)$ and $Enc(\varphi_2)$ can also be shown to be 1-WL indistinguishable. This can be observed intuitively, as node neighborhoods in both graphs are identical and very regular: all variable nodes are connected to exactly one other variable node and two disjunction nodes, and all disjunction nodes are connected to exactly two variables.

### A.3.2 CONSTRUCTION OF CEXP

Given a EXP dataset with $N$ pairs of graphs, we create CEXP by selecting $N/2$ graph pairs and modifying them to yield CORRUPT. The unmodified graph pairs are therefore exactly identical in type to EXP instances, and we refer to these instances within CEXP as $\overline{\text{EXP}}$.

For every graph pair, we discard the satisfiable graph and construct a new graph from a copy of the unsatisfiable graph as follows.

1. Randomly introduce new literals to the existing disjunctions of the copy of the unsatisfiable graph, such that no redundancies are created (i.e., adding $x$ to a disjunction when $x$ or $\bar{x}$ is already present), until 3 literals are added *and* the formula becomes satisfiable. Literal addition is done by creating new edges in the graph between disjunction and literal nodes. To do this, disjunctions with less than 5 literals are uniformly randomly selected, and the literal to add is uniformly randomly sampled from the set of all non-redundant literals given the selected disjunction.

2. Once a satisfiable formula is reached, iterate sequentially over all added edges, and eliminate any edge whose removal does not restore unsatisfiability. This ensures that a minimal number of new edges, relative to the original unsatisfiable graph, are added.

Observe that these modifications have several interesting effects on the dataset. First, they preserve the existing UNSAT core nodes and edges, while flipping the satisfiability of their overall formulas, which makes the learning task go beyond structure identification. Second, they introduce significant new variability to the dataset, in that the planar component and cores can share edges. Finally, they make the graph pairs 1-WL distinguishable, which gives standard GNNs a chance to perform well on CORRUPT.

### A.3.3 DATASET GENERATION FOR EXPERIMENTS

To create the EXP dataset, we randomly generate 600 core pairs, where $n$ (cf. Appendix A.3) is uniformly randomly set between 2 and 4 inclusive. Then, we generate the additional planar component using Plantri, such that 500 $\varphi_{\text{planar}}$ formulas are generated from 12-node planar bipartite planar graphs, and the remaining 100 from planar bipartite graphs with 15 nodes.

This generation process implies that every formula has a number of variables ranging between 10 (4 core variables when $n = 2$ plus a minimum 6 variables from the larger bipartite set during $\varphi_{\text{planar}}$ generation from 12-node graphs) and 22 variables (8 core variables for $n = 4$ plus a maximally-sized variable subset of 14 nodes for $\varphi_{\text{planar}}$ generation from 15-node graphs).

Furthermore, the number of disjunctions also ranges from 10 (8 core disjunctions for $n = 2$ plus the minimum 2 disjunctions for the case where $\varphi_{\text{planar}}$, generated from 12-node graphs, has 10 variables and 2 disjunctions) to 30 disjunctions (16 core disjunctions for $n = 4$ plus at most 14 disjunctions for the case where $\varphi_{\text{planar}}$, generated from 15-node graphs, initially has 8 variables and 7 disjunctions, which can at most lead to 14 final disjunctions following step (ii)).

### A.4 STANDARD DEVIATION OF GCN-50%RNI ON EXP OVER TRAINING

In this subsection, we investigate the variability of GCN-RNI learning across validation folds, and do so with a representative model and dataset, namely the semi-randomized GCN-50%RNI model and the standard EXP dataset. The standard deviation of the test accuracy of GCN-50%RNI over EXP, across all 10 cross-validation folds relative to the number of epochs, is shown in Figure 5. From this figure, we see that standard deviation spikes sharply at the start of training, and only begins dropping after 100 epochs. This suggests that the learning behavior of GCN-50% RNI is quite variable, sometimes requiring few epochs to converge, and in other cases requiring a very high number of epochs. Furthermore, standard deviation converges to almost zero following 200 epochs, corresponding to the phase where all

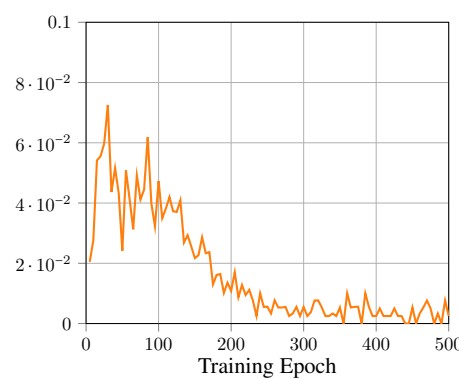

Figure 5: Standard deviation of test accuracy over all 10 validation splits of GCN-50%RNI on EXP.

validation folds have achieved near-perfect test performance. From these findings, we further confirm that RNI introduces volatility to GCN training, this time manifesting in variable convergence times across validation folds, but that this volatility does not ultimately hinder convergence and performance, as all folds eventually reach satisfactory performance within a reasonable amount of epochs, and subsequently stabilize at this level.

### A.5 ADDITIONAL EXPERIMENTS

In addition to the experiments in the main body of the paper, we additionally evaluate RNI on sparser analog datasets to EXP and CEXP, namely SPARSEEXP and SPARSECEXP. These datasets only contain 25% of the number of instances of their original counterparts, and are used to study the behavior and impact of RNI when data is sparse.

### A.5.1 EXPERIMENT 1 ON SPARSEEXP

In this experiment, we generate SPARSEEXP analogously to EXP, except that this dataset only consists of 150 graph pairs, i.e., 300 graphs in total. We then train 3-GCN for 200 epochs, and all other systems for 1000 epochs on SPARSEEXP, as opposed to 100 and 500 respectively for EXP, to give all evaluated models a better opportunity to compensate for the smaller dataset size. We show the learning curves for all models on SPARSEEXP, and reproduce the original figure for EXP, in Figure 6 for easier comparison.

First, we observe that all models converge slower on SPARSEEXP compared to EXP. This is not surprising, as a lower data availability makes learning a well-performing function slower and more challenging. More specifically, sparsity implies that (i) fewer weight updates are made per epoch, and (ii) these updates are of lower quality, as they are computed from a less representative and complete dataset. Nonetheless, the same relative convergence patterns between GCN-RNI models and 3-GCN are also visible in this setting, further highlighting the increased convergence time required by GCN-RNI models.

We also observe that all GCN-RNI models, though also eventually converging, do so in a more volatile fashion. Indeed, GCN-RNI models suffer from the sparseness of the dataset, as this makes them more sensitive to RNI. As a result, these models require more training to effectively learn robustness against RNI values, and learn this from a smaller sample set, increasing their variability further. Moreover, the nature of SPARSEEXP makes learning more difficult, as it fully relies on RNI for MPNNs to have a chance of achieving above-random performance, and thus encourages MPNNs to fit specific RNI values. Hence, RNI introduces significant volatility and variability to training, particularly with sparser data, and requires substantial training and epochs for GCN-RNI models to effectively develop a robustness to RNI instantiations.

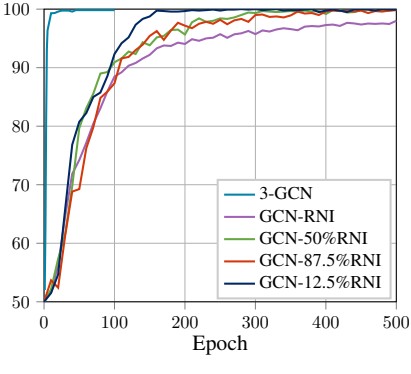

(a) Learning curves on EXP.

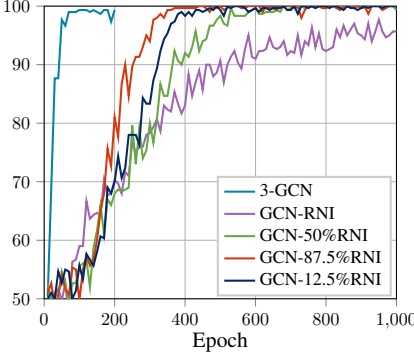

(b) Learning curves on SPARSEEXP.

Figure 6: Model convergence results for Experiment 1 on the datasets EXP and SPARSEEXP.

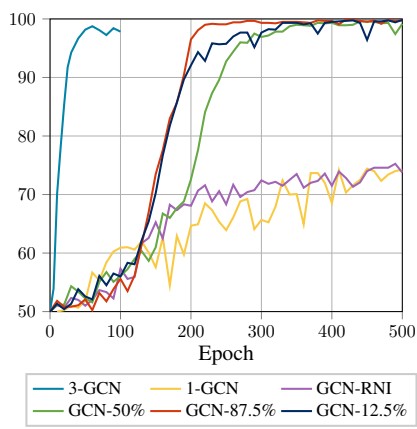

(a) Learning curves for all GCN-RNI models and 3-GCN on CEXP.

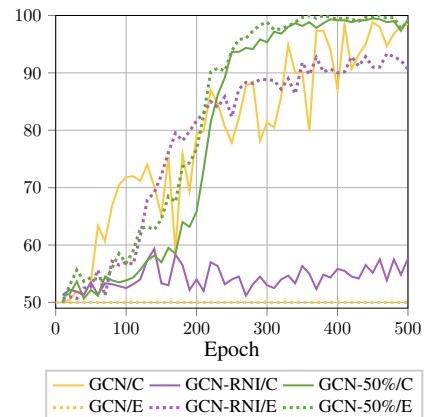

(b) Learning curves for CEXP, split across $\overline{\text{EXP}}$ (/E) and CORRUPT (/C).

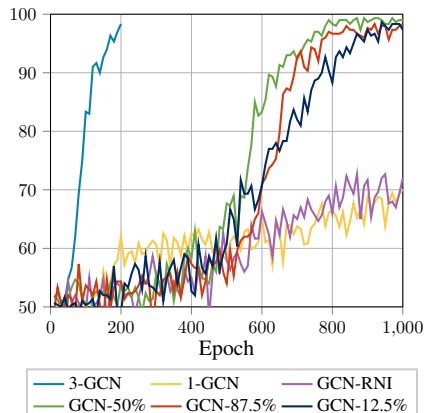

(c) Learning curves for all GCN-RNI models and 3-GCN on SPARSECEXP.

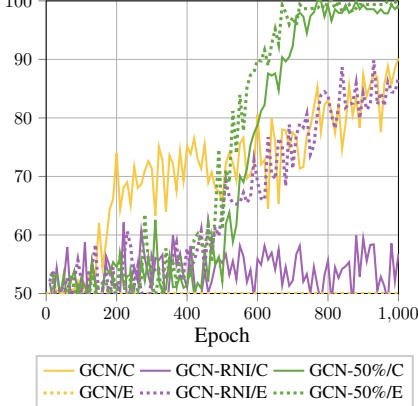

(d) Learning curves for SPARSECEXP, split across $\overline{\text{EXP}}$ (/E) and CORRUPT (/C).

Figure 7: Model convergence results for Experiment 2 on CEXP and SPARSECEXP.

### A.5.2 EXPERIMENT 2 ON SPARSECEXP

Analogously to Experiment 1, we generate a SPARSECEXP dataset similarly to CEXP, but only generate 150 graph pairs. Then, we select 75 graph pairs and modify them, as described in Appendix A.3.2. We report the learning curves for all models on SPARSECEXP, as well as the original curves for CEXP from the main body, in Figure 7.

Table 2: Hyper-parameter configurations for all GCN-RNIand 3-GCN experiments.

| Dataset | EXP | | CEXP | |
|---|---|---|---|---|
| | $\lambda$ | $p$ | $\lambda$ | $p$ |
| GCN | $1 \times 10^{-4}$ | N/A | $1 \times 10^{-4}$ | N/A |
| GCN-12.5%RNI | $2 \times 10^{-4}$ | N | $2 \times 10^{-4}$ | N |
| GCN-50%RNI | $2 \times 10^{-4}$ | N | $2 \times 10^{-4}$ | N |
| GCN-87.5%RNI | $2 \times 10^{-4}$ | N | $5 \times 10^{-4}$ | N |
| GCN-RNI | $5 \times 10^{-4}$ | N | $5 \times 10^{-4}$ | N |
| 3-GCN | $5 \times 10^{-4}$ | N/A | $2 \times 10^{-4}$ | N/A |

Table 3: Performance of GCN-RNI models on the EXP dataset with the hyperbolic tangent activation function.

| Model | Testing Accuracy (%) |
|---|---|
| GCN-RNI(U) | $92.7 \pm 5.61$ |
| **GCN-RNI(N)** | **$96.0 \pm 2.11$** |
| GCN-RNI(XU) | $64.6 \pm 19.9$ |
| GCN-RNI(XN) | $63.0 \pm 20.9$ |

As in the previous subsection, similar behavior is observed on SPARSECEXP compared with CEXP, only differing by slower convergence in the former case. However, we note that the "struggle" phase described in the main paper, which only occurs during the first 100 epochs over CEXP, lasts for around 500 epochs on SPARSEEXP. Intuitively, this "struggle" phenomenon is due to conflicting learning requirements, stemming from CORRUPT and $\overline{\text{EXP}}$, which effectively require models to "isolate" deterministic dimensions for CORRUPT, and other randomized dimensions for $\overline{\text{EXP}}$. This in itself is already challenging on CEXP, but is made even more difficult on SPARSECEXP due to its sparsity. Indeed, sparsity makes that further samples are needed in expectation to find a reasonable solution, leading to a lengthy "struggle" phase, in which both CORRUPT and $\overline{\text{EXP}}$ data points conflict with one another during optimization.

### A.6  HYPER-PARAMETER DETAILS

All GCN models with (partially or completely) deterministic initial node embeddings map a 2-dimensional one-hot encoding of node type (literal or disjunction) to a $k$-dimensional embedding space, where $k$ corresponds to the dimensionality of the deterministic embeddings. Furthermore, the final prediction for every graph is computed by aggregating all node embeddings following message passing using the max function, and then passing the result through a multi-layer perceptron of 3 layers with dimensionality $x$, 32 and 2 respectively, where $x$ is the embedding dimensionality used in the given model. The activation function for the first two MLP layers is the ELU function (Clevert et al., 2016), and the softmax function is used to make a final prediction at the final MLP layer.

All neural networks in this work are optimized using the Adam optimizer (Kingma & Ba, 2015). All training is conducted with a fixed learning rate $\lambda$, for fairer comparison between all models. Initially, decaying learning rates were used, but these were discarded, as they yielded sub-optimal convergence for all GCN-RNI models. Finally, all experiments were run on a V100 GPU. Detailed hyper-parameters, namely learning rate $\lambda$ and RNI distribution $p$, per model on every evaluation dataset are shown in Table 2.

### A.6.1  RESULTS FOR GCN-RNI WITH HYPERBOLIC TANGENT ACTIVATION

In addition to experimenting with the RNI probability distribution, we also experimented with different activation functions for the GCN message passing iterations. Results are shown in Table 3. Performance with $tanh$ is significantly more variable across distributions than ELU, which shows that RNI can be highly sensitive to practical choices of hyper-parameters.

