# OpenReview forum: "The Surprising Power of Graph Neural Networks with Random Node Initialization"
_ICLR.cc/2021/Conference — Reject_

### Official Review · AnonReviewer1 · 2020-10-27
**Why does randomness help?**

**Rating:** 5
**Confidence:** 4

**Review:**

The paper studies the how random initialization of node states can improve the expressivity of message passing graph neural networks. Theoretically the paper shows that RNI makes MPNNs universal approximators for invariant functions over graphs. To supplement this claim, the authors evaluate GNNs with RNI and higher-order GNNs over a carefully constructed synthetic dataset and show that RNI (even if only a fraction of the nodes are randomly initialized) are as expressive as higher order GNNs. They also highlight some drawbacks of this approach, such as slower training and sensitivity to hyperparameter tuning.

Overall, I find the topic and the findings of the paper quite interesting. However, explanations as to why or how randomness helps is missing in the paper. There is also room for improvement in the presentation and writing. The statement of Theorem 4.1 is a little vague; it would help if its made more precise (e.g., how much randomness is required, what is the depth of the MPNN required, the state dimensions etc.). The notion of invariance and equivariance can also be defined much earlier in the paper.

While it’s interesting that randomness can yield MPNNs to be universal approximators, could you please give some intuition as to what makes randomness so essential? In the proof of Theorem 4.1 (in Appendix A.2), the random initialization together with the linearized sigmoid activation function is used to show that the vectors $x_i^(1)$ are mutually distinct {0, 1} vectors with high probability. To my understanding, this is the only place where the randomness of the initialization is used. If that is the case, why can’t any deterministic initialization that ensures mutually distinct {0, 1} vectors for $x_i^(1)$ suffice?

The datasets EXP and CEXP seem to be motivated as graph encoding of SAT problems. Are they bipartite graphs? Could you give a sense for how the graphs look like (number of nodes, edges etc.)?

In the partial random node initialization (GCN-x% RNI), are you initializing some nodes randomly and other deterministically? Or are you initializing some entries in the state vector randomly and remaining entries deterministically?

---

> ### Author Response · Authors · 2020-11-12
> **Author Response**
>
> We thank the reviewer for the constructive feedback, and address the concerns below:
>
> - **Theorem:** We now explicitly mention the required theoretical width, shown in the appendix, in the main body of the paper (Page 5).  In terms of GNN depth, our result has the same required depth as that of Barcelo et al. We also now provide intuitions, and a proof sketch to better clarify the role that RNI plays after the theorem statement (Page 5). The intuition behind our proof is that RNI can implicitly induce a total order over graph nodes, which in turn, makes these graphs distinguishable. Since this representation is produced with a probability $(1 - \delta)$, we obtain the approximability result given in the paper. Please also refer to our answer to Reviewers 4 and 2.
>
> - **Why not deterministic extra features?** The fundamental motivation for randomization over deterministic extra features is generalizability. More specifically, adding deterministic distinguishing features introduces redundant information which plays no part in prediction, and this itself leads to overfitting when training a GNN. Indeed, we have experimented with such a deterministic setup, where we set the same added features for every graph (these were generated randomly, but were preserved for every run), but this yielded very poor performance, albeit marginally better than 50% in terms of accuracy. By contrast, the results that we report with repeated randomization show near-perfect performance, and we believe this is due to the GNN not fitting to the randomized dimensions due to their variability, and thus developing a robustness to these added features, which is not possible with deterministic features.
>
> - **The properties of the datasets EXP/CEXP:** As both datasets consist of SAT formulas, they consist of nodes representing literals and disjunctions, such that literals that are negations of one another are connected, and disjunction nodes are connected to literal nodes for all literals that appear in the disjunction. Hence, although CEXP/EXP are planar, they are not bipartite, due the inter-literal edges. In terms of #nodes, #edges, the instance sizes are small and typically do not exceed 70 nodes, and the number of edges would be roughly the number of instance variables (12-16) plus disjunction connections (at most 5 per disjunction by construction). Further details of these datasets, including the generation process, are provided in the appendix of the paper.
>
> - **Partial RNI:** Our partial randomization is over node features, and not nodes themselves, so the latter option. That is, partial randomization implies that every node only has a fraction of its dimensions randomized (as is the case in Sato et al.), and not that a subset of nodes is fully randomized. We acknowledge that this was ambiguously written in the paper, and have clarified this in the updated version.
>
>
> [1] Sato et al. "Random Features Strengthen Graph Neural Networks." arXiv 2002.03155
> [2] Barcelo' et al. "The Logical Expressiveness of Graph Neural Networks", ICLR 2020.

---

### Official Review · AnonReviewer3 · 2020-10-28

**Rating:** 5
**Confidence:** 4

**Review:**

This paper studies the power of message passing neural networks (MPNNs) with random node initialization (RNI). Although the power of standard MPNNs is limited to 1-WL, the main result of the paper is to prove that RNI makes MPNNs universal. The paper also introduces two graph classification datasets where each graph is a SAT problem and the label is the satisfiability/unsatisfiability. The datasets have been created in a way that their graphs are 1-WL indistinguishable to serve as a test-bed for graph neural networks with power beyond 1-WL. The results on these two datasets show the merit of adding RNI to MPNNs.

My biggest reservation with this work is that the theoretical and empirical results do not seem to offer much more than what has already been provided in [1]. In fact, if [1] didn't exist, I would have given a strong acceptance to this paper. I’ll elaborate on each of these aspects below:

Theoretical result: While Theorem 4.1 is quite strong as it imposes no restrictions on the function f, two things are not clear to me: 1- What classes of functions does the Theorem 1 of [1] not cover that Theorem 4.1 of this paper does?, 2- Is the Theorem 4.1 of this paper a straightforward extension of the Theorem 1 in [1]?

Empirical results: While I appreciate the two datasets developed in this work and I believe they can be useful for future research, I have a hard time understanding what intuition/insights the results provide that has not been already provided in [1]. It has been already shown in [1] that RNI increases the power of MPNNs and enables them to do well on graphs that are 1-WL indistinguishable, where existing MPNNs fail. If Theorem 4.1 covers a larger class of models than Theorem 1 of [1], then I would expect at least some experiments covering those classes of models. Right now, the only added insight of the empirical results seems to be that randomizing a subset of the nodes is better than randomizing all of them.

All mentioned about the connection between this work and [1] is “Indeed, RNI has enabled MPNNs to distinguish instances that 1-WL cannot distinguish, and is proven to enable better approximation of a class of combinatorial problems (Sato et al., 2020). However, the effect of RNI on the expressive power of GNNs has not yet been comprehensively studied, and its impact on the inductive capacity and learning ability of GNNs remains unclear.”, which does not seem satisfactory.

Other comments/questions: 1- Some experiments on standard benchmarks can strengthen the paper. 2- For GCN-x%RNI, how do you initialize the other (1-x)% of the nodes? Do you initialize them as all 0s? 3- In [1], a new random feature is assigned every time the procedure is called. Is this what you do as well? 4- Any reason why the results of 1-GCN and GCN-x%RNI are not reported in Table 1?

[1] Sato, Ryoma, Makoto Yamada, and Hisashi Kashima. "Random Features Strengthen Graph Neural Networks." arXiv preprint arXiv:2002.03155 (2020).

---

> ### Author Response · Authors · 2020-11-12
> **Author Response**
>
> We thank the reviewer for the comments, and address the main points below:
>
> **Contribution relative to Sato et al.:** Please note that the question of theoretically quantifying the impact of RNI on GNN expressiveness remained open; quoting from Sato et al.:
>
>   (i) "Although this heuristic seems to work well, it is not trivial whether adding a random feature can theoretically strengthen the capability of GNNs", and
>
>   (ii) "We prove that the addition of random features indeed improves the theoretical capability of GNNs *in terms of the approximation ratios*".
>
> Sato et al. only addresses the theoretical power of RNI in the limited context of approximation ratios for some combinatorial problems, based on analogies with existing algorithms, which have known approximations, showing that RNI can help detect local structures in an input graph with high probability. This clearly leaves several questions open, including all functions beyond substructure detection and all functions which don't have an approximation. Our result shows the universality of GNNs with RNI in the general case, and is hence a strict generalisation of Sato et al.
>
> **Responses to the specific questions:**
>
> (1) Sato et al. only addresses the theoretical power of RNI for some combinatorial problems, and anything beyond is not captured. These results are algorithmic (in the sense of algorithmic alignment), and hence limited to functions which have a known approximation. The problem used in our experiments, i.e., SAT solving is one example that is not captured by Sato et al. Indeed, SAT is NP-complete even on our planar instances, is hard to approximate, and does not rely on fixed local structures. Essentially, our result addresses any real-valued function over graphs, and shows it to be theoretically learnable, and is much broader, and provides a more complete picture on the role of RNI.
>
> (2) Our result is by no means a straightforward extension to Theorem 1 of Sato et al., as it proves a much more general statement. We prove a universality result, based on a logical characterization of the power of GNNs with RNI with a descriptive complexity argument. By contrast, Sato et al. builds on specific problem settings, and aligns its GNN models with the specific problem setting to yield its results, so their approach, cannot be extended to yield universality, as several problems, either 1) do not admit approximation results to be adapted into the settings, or 2) do not rely on fixed local substructures, or both.
>
> **Empirical results compared to Sato et al.:** Our datasets are based on the SAT problem, where neither local structures are useful nor approximation schemes are available. Hence, our experiments evaluate GNNs with RNI on a significantly more challenging problem setting. We also address MPNNs, but we also go beyond this and compare with more powerful models, namely 3-GCNs, to obtain better insights on the practical gains stemming from RNI. Hence, we provide several empirical insights in addition to the insights given by partial randomization that, which we summarize below:
>
>   (a) Going beyond Sato et al, we show that RNI is also beneficial in a challenging setting not amenable to approximation or local structures.
>
>   (b) We compare MPNNs with RNI with higher-order GNNs to better quantify the impact of RNI, and show that MPNNs with RNI are competitive with these-order models. That is, we don't only show that MPNNs with RNI outperform MPNNs, but also that MPNNs with RNI can even surpass higher-order GNN models. We are not aware of any work that quantifies the performance of higher-order GNN models on datasets specifically dedicated to test expressivity.
>
>   (c) We conduct a careful study based on convergence of different models, and illustrate that randomization slows down convergence. Additionally, we give concrete evidence that a partial randomization regime can alleviate this problem.
>
> We made all these points clearer in the related work section (Page 4), to better highlight the stronger nature of our results.
>
> **Other comments:**
>
>   (1) Sato et al. perform such a study, and show the benefit of randomization, using an analogous randomization regime as our own, on real-world datasets.
>   (2) Partial RNI is at node feature level, as in Sato et al., and not at node level. We have addressed this ambiguity in the updated version of the paper.
>   (3) Yes, we follow the same randomization strategy.
>   (4)The result of 1-GCN is 50%, as it cannot distinguish between EXP graph pairs, whereas the results of partial RNI systems are highly similar, and even slightly better, than those of full RNI, and are plotted in Figure 2.

---

> > ### Comment · AnonReviewer3 · 2020-11-22
> > **Response**
> >
> > Thanks for the response.
> >
> > The provided response seems to disparage the contributions of Sato et al.
> > It quotes "Although this heuristic seems to work well, it is not trivial whether adding a random feature can theoretically strengthen the capability of GNNs” as evidence that Sato et al. have left the impact of RNI on GNN expressiveness as an open problem. But this is wrong because that sentence only corresponds to a motivating example that Sato et al. provide, which they then used as a reason to study the theoretical expressiveness of GNNs with RNI in their section 4.3 and Theorem 1.
> >
> > The response also quotes "We prove that the addition of random features indeed improves the theoretical capability of GNNs in terms of the approximation ratios” as evidence that the work of Sato et al. only considers (quoting from the response) "the limited context of approximation ratios”. While the quoted sentence is indeed coming from Sato et al., it does not mean that they only studied the theoretical capability of GNNs in terms of approximate ratios. In their section 4.3, they study the "Expressive Power of rGINs” (which is the same problem studied in this submission and goes beyond approximate ratios) and mention "we prove that rGINs can distinguish any local structure w.h.p.”.
> >
> > For the same reason as above, the quotes "Sato et al. only addresses the theoretical power of RNI for some combinatorial problems” from the response do not seem to be accurate either, and they disparage the contributions of Sato et al.
> >
> > The authors claim that SAT is a different and significantly more challenging problem than those considered in the work of Sato et al because local structures are not useful in SAT solving. From what I understand, to solve a SAT problem, one can add an auxiliary node to the graph and connect it to every other node (this is a well-known approach for graph classification) and then the SAT solving problem reduces to distinguishing the local structure around the auxiliary nodes (which is captured by Theorem 1 of Sato et al.). Isn’t this correct?

---

> > > ### Author Response · Authors · 2020-11-22
> > > **Author Response (Part 1)**
> > >
> > > Thank you very much for replying during the rebuttal period. This allows us to further clarify our points, and we very much appreciate this.
> > >
> > >
> > > > “The provided response seems to disparage the contributions of Sato et al.”
> > >
> > > We absolutely do not intend to belittle the contributions of Sato et al, and in fact we developed our work to complement these findings, and credit their findings adequately: Sato et al. shows that rGNN enables the detection of arbitrary (sub)structures with high probability (Theorem 1) and addresses the ability of rGIN with respect to approximating multiple combinatorial problems (Section 4.4). Please note that this leaves the case of learning arbitrary functions, not involving a fixed structure, open. This is what motivated our (more comprehensive) study of the power of rGNN, leading to a general universality result that fills this gap with a unifying argument.
> > >
> > > We are well aware of Theorem 1, and have already mentioned its impacts in our initial response, “showing that RNI can help detect local structures in an input graph with high probability”. It is correct that we have highlighted the study of combinatorial problems and approximation ratios more, but we did so, since we see that these are the major contributions of Sato el al.  Indeed, Sato et al. themselves do not claim universality (from Theorem 1), hence we prioritized the latter part of their contribution in our comparison.  We now understand and acknowledge that the differences between our main result and Theorem 1 of Sato et al should have been more clearly stated in our response. Therefore, we will provide a detailed comparison of our result with Theorem 1:
> > >
> > >
> > > - In Sato et al., Theorem 1 states that, given any graph $G’$, corresponding to a fixed structure, there exists a parametrisation $\theta$ of rGIN, such that, for all graphs $G \in \mathcal{G}$, if $G’$ (up to isomorphism) is in $G$, then rGIN returns a positive score w.h.p. Otherwise, rGIN returns a negative score w.h.p. Thus, Theorem 1 shows that rGIN parametrizations are capable of detecting a (sub)structure in input graphs w.h.p. and doesn't have implications beyond this.
> > >
> > > - Theorem 1 does *not* imply universality, and neither is it claimed as such. In particular, Theorem 1 states that an rGIN parametrization $\theta$ exists to detect a unique, fixed structure $G’$, but this does not show that this parametrization can jointly detect multiple structures. Indeed, for a function defined over sets of graphs $G_1, … G_n$, which do not share a relevant substructure, Theorem 1 only shows that $n$ potentially different parametrizations of rGINs exist, each of which corresponds to detecting $G_1, … G_n$ individually, but not that a single parametrization exists to jointly capture all these graphs. Hence, the theorem does not show that rGIN can capture arbitrary functions over graphs, and so does not show universality. By contrast, our result shows the universality of MPNNs with RNI, namely that arbitrary Boolean (and real-valued) functions over graphs can be captured by these models. Thus, our result strictly and substantially generalizes that of Sato et al. to the arbitrary function setting.

---

> > > > ### Author Response · Authors · 2020-11-22
> > > > **Author Response (Part 2)**
> > > >
> > > > > But this is wrong because that sentence only corresponds to a motivating example that Sato et al. provide
> > > >
> > > > Indeed, Sato et al. use this statement to motivate their example and subsequent theoretical expressiveness study in Section 4.3, as you highlight, but this study is not complete, as it leaves the aforementioned general case open. Hence, the non-triviality of theoretically studying the expressiveness of rGNNs remains a valid statement in the complete sense of GNN expressiveness, and thus we do not think that we misquote this statement.
> > > >
> > > > > The response also quotes "We prove that the addition of random features indeed improves the theoretical capability of GNNs in terms of the approximation ratios” as evidence that the work of Sato et al. only considers (quoting from the response) "the limited context of approximation ratios”. While the quoted sentence is indeed coming from Sato et al., it does not mean that they only studied the theoretical capability of GNNs in terms of approximate ratios. In their section 4.3, they study the "Expressive Power of rGINs” (which is the same problem studied in this submission and goes beyond approximate ratios) and mention "we prove that rGINs can distinguish any local structure w.h.p.”
> > > >
> > > > As mentioned earlier, it was never our intention to appear dismissive of Theorem 1, or to solely highlight Section 4.4. We have mentioned these in our initial response, but do acknowledge this could have been more detailed. Nonetheless, we hope that this response now makes our contribution, as well as our understanding of Sato et al.’s results, more pronounced, unambiguous and explicit.
> > > >
> > > > > For the same reason as above, the quotes "Sato et al. only addresses the theoretical power of RNI for some combinatorial problems” from the response do not seem to be accurate either, and they disparage the contributions of Sato et al.
> > > >
> > > > We understand how this statement could have caused confusion. Indeed, Sato et al. further learns functions detecting fixed local substructures following Theorem 1, and we were aware of this when writing our answer: “…and does not rely on fixed local structures” (referring to Theorem 1), but we understand why this was misleading. Once again, we have no intention to disparage the contribution of Sato et al., but in fact to build on it. We thank the reviewer for highlighting this ambiguity.
> > > >
> > > > > From what I understand, to solve a SAT problem, one can add an auxiliary node to the graph and connect it to every other node (this is a well-known approach for graph classification) and then the SAT solving problem reduces to distinguishing the local structure around the auxiliary nodes (which is captured by Theorem 1 of Sato et al.). Isn’t this correct?
> > > >
> > > > This is not correct, as the set of structures around such an auxiliary node, can be arbitrarily large (and thus trivially non-unique), and the absence/presence of a single given structure (e.g., triangle, bipartite subgraph) cannot solely determine the satisfiability of a formula. This means that Theorem 1 cannot apply in this setting. More concretely, Theorem 1 only shows the existence of a parametrisation of rGIN that detects a single local structure, which clearly is not sufficient to solve SAT, as SAT (1) does not have any fixed determining structures that dictate the satisfiability of formulas, as we argue in our first response,  and (2) would surely require, at the very least, the joint detection of several distinct structures, which is not given by Theorem 1. Please allow us to restate that, based on the SAT problem, we also make a principled comparison against higher-order models, which is not present in any existing work thus far.
> > > >
> > > > We hope that this response clarifies your concerns.  Please let us know and we would be happy to follow up.
> > > > Thanks!

---

### Official Review · AnonReviewer2 · 2020-10-28
**Good paper and contribution.**

**Rating:** 7
**Confidence:** 3

**Review:**

**Post Rebuttal**
I thank the authors for the quick replies and updates to the paper.
I keep my positive score.

---


**Summary of Contributions**
The paper analyzes the model of Random features in GNNs as suggested by Sato et.al., in the paper called RNI.
A result proving the universality of the RNI framework is introduced, a first of its kind in low tensor degree GNNs.
To evaluate the expressiveness of RNI and other more expressive GNNs, the authors design two datasets wich require 2-WL distinguishing power (which is higher than the ones MPNNs have)

**Strengths**
- Novelty - The universality result on RNI is novel and further extends the hints of improved expressiveness explored by Sato et.al.
- Dataset design - the design of new datasets for expressiveness discrimination are an important contribution to the community.

**Weaknesses**

- Regarding invariance of RNI - a more rigorous explanation would be fit there. Why does RNI preserve invariance?
- The main theoretical result of the paper is just appearing in the paper without details and intuitions towards the proof, a proof sketch or some discussion of that flavor would make the result more clear.
- The experimental setting is not clear enough as presented in the main body of the paper. A lot of important details has to be dug our from the appendix which is tedious for the reader. For example:
1. The partial RNI is not well explained in the paper, and it was not clear whether the partial applies to the feature dimensions or the nodes.
2.  The input features in the designed datasets are not mentioned in the body of the paper.
3. An elaborate description of the 3-GCN variant is missing.

- *Partial RNI* - can the authors provide an intuition as to why it works? In a way, the universality comes from the network not taking into account the node features due to the full randomness but more of a statistical behavior and the fact that the nodes are completely distinguishable. So why does partial randomness work?


**Recommendation**
The paper states an important and surprising result which can contribute greatly to the graph learning community.
A good paper, Accept.

---

> ### Author Response · Authors · 2020-11-12
> **Author Response**
>
> We thank the reviewer for the constructive feedback, and respond to the comments below:
>
> - **Invariance of RNI:** RNI introduces continuous node features which are re-sampled at every iteration of training or evaluation. These features, by construction, vary around a mean value and, in expectation, this mean value is what the GNN will rely on for predictions. However, the variability between different samples, and in particular that of a random sample relative to its mean, enables graph discrimination and improves expressiveness. Hence, in expectation, RNI preserves invariance, as all samples ultimately converge on average to a single value that can be sampled, whereas variance is leveraged for increased expressiveness. We clarified this further in the highlighted part of the paragraph preceding Theorem 4.1 (Page 4).
>
> - **Proof Intuitions:** We have provided additional intuitions for the proof in the main body of the paper with a new paragraph in Page 5 after Theorem 4.1. Our result extends  Barcelo et al.'s result, showing that the class of functions captured by MPNNs (or ACR-GNNs) are precisely those in $\text{C}^2$. The intuition behind our proof is to use RNI in order to create an implicit total order over graph nodes, and make use of the fact that such ordered graphs can always be distinguished. The ordered representation is produced with a probability $(1 - \delta)$, where $\delta$ affects embedding width, and based on this representation, we show that any function over graphs can be learned, by proving the result for the class of Boolean functions, and then lifting them to the general case.
>
> - **Experimental Setup:** Thanks for pointing this out:
>     (1) Partial RNI: We acknowledge that this is not clearly stated in the paper, and have explicitly and unambiguously mentioned this in the paper, particularly in the experimental setup section presenting GCN-xRNI (Page 6). The partial RNI indeed applies to feature dimensions, i.e., a 50-dimensional node embedding with 50%RNI has 25 random dimensions, and 25 deterministic dimensions, as has also been done in Sato et al.
>     (2) Features: We also explained the features in the experimental setup (Page 6), as part of the GCN-xRNI description.
>     (3) 3-GCN description: This was shortened in the submitted version due to space constraints. We have now extended the model description in the experimental setup.
>
> - **Partial RNI Intuition:** Let us start by noting that our universality result also works with partial RNI. In fact, it already holds with only *one* randomized dimension (now mentioned in Page 5). That is, additional deterministic features do not affect the universality result as long as there is a single randomized dimension, since the deterministic dimensions can be simply concatenated without changing any of the derived probabilities in Theorem 4.1. So, it is theoretically quite plausible for partial RNI to do well. The fact that it tends to do better in practice is discussed in the experimental section the paper: Partial RNI somehow achieves a 'best-of-both-worlds' scenario, where the GNN has access to both deterministic node features (with better inductive bias) which are informative for the prediction task, and random features to improve its expressiveness. Practically, partial RNI achieves the same benefits as full RNI, in that it enables node distinguishability, and this highlights that little randomization is practically needed to improve MPNNs.  Hence, partial RNI supplements the power of random features with informative deterministic features, which lends itself to better prediction performance, particularly on CEXP.
>
> [1] Barcelo' et al. "The Logical Expressiveness of Graph Neural Networks", ICLR 2020.
> [2] Sato et al. "Random Features Strengthen Graph Neural Networks.", arXiv 2002.03155

---

### Official Review · AnonReviewer4 · 2020-10-28
**Well written paper about random features in GNNs**

**Rating:** 7
**Confidence:** 3

**Review:**

The paper study the effects of adding random features (RF) to graph neural networks (GNN). First, it is shown that, quite surprisingly, adding random features makes GNN universal approximators of invariant functions. Next, a novel dataset is defined that is aimed at evaluating the performance of models that have high expressive power. Finally, several experiments show that adding RF performs well on the proposed dataset.

I think the paper is well written and well organized. The theoretical aspect seems novel and quite surprising, especially since it shows that adding RF makes GNN more expressive than k-GCN (or k-IGN) for any k. The new proposed dataset is interesting, and the experimental result looks promising, especially since RF performs quite well compared to 3-GCN while having much fewer parameters. Also, the observation that only partial randomness can already be beneficial is interesting.

I have a couple of concerns regarding the paper for which I would be happy to see the authors’ comment:
1)	The phrasing of Theorem 4.1 is a bit vague because it is stated with an \epsilon, \delta approximation while there is no clear explanation of how these parameters affect the theorem. I suppose that smaller \epsilon, \delta would mean a wider network, but this dependence should be shown explicitly. For \delta, this dependence is somewhat shown in the proof of Lemma A.5, but for \epsilon it is very unclear and is probably related to Lemma A.2 which cited from another paper.
2)	If I understand the proof correctly, the width of the network in Lemma A.1 (and also in Theorem 4.1) should be super-exponential, no matter what is the target function. That is because in Lemma A.4 the sentence takes into account every graph in G_{n,k}, the number of such graphs is super-exponential and I suppose that the width of the GNN that realizes the sentence depends on the length of the sentence (or at least the number of literals). If this is true, then I think it is important to point that out in the main text.
3)	Continuing the previous point, the paragraph after Remark 1 isn’t clear: why the construction is adaptive to the descriptive complexity of the target function if for any target function the construction requires memorization of all graphs with n nodes?
4)	Regarding the Experimental results, I think it is important to also test RF on real datasets. The reason is that without this experiment it is not clear if adding RF doesn’t actually harm the performance of GNN and thus are actually very impractical. As the authors stated, I don’t expect RF to significantly improve performance on real datasets because high expressivity might not be required, but I am concerned that it will hurt the performance.

I also highly suggest uploading the EXP and CEXP datasets as supplementary material, since these are newly generated datasets, and this way other people could also experiment on them and compare performance.

To conclude, I think this is a very nice and well-written paper that adds a novel view on adding RF to GNN, both theoretically and empirically. With that said, there are some issues with the vague form in which the main theorem is stated, and experiments on real datasets would help clarify whether adding RF could actually prove helpful (or at least not harmful) for practical uses.

---

> ### Author Response · Authors · 2020-11-12
> **Author Response**
>
> We thank the reviewer for the insightful feedback and points. We are very pleased that you find our paper interesting: Our results indeed support the viability of RF as an efficient alternative to higher-order models. We address your concerns below:
>
> 1. Lemma A.1 (Boolean functions) and Theorem 4.1 (general functions) have the following implications:
>
>     (i) $\delta$: The confidence parameter $\delta$ affects embedding dimensionality (i.e., GNN width): In the proof of Lemma A.1, we show that the needed dimensionality is $O(n^2 \delta^{-1})$, where n is the maximal number of graph nodes. Therefore, it is correct that lower $\delta$ implies higher GNN width.
>
>     (ii) $\epsilon$: The error $\epsilon$ is only used in defining a universal readout function, and is needed in our (and, in Barcelo et al.'s) construction solely for this reason. Specifically, it thresholds the error of the GNN when making approximations of the target function, defining a universal readout function (e.g., MLP) following the GNN to enable $\epsilon$ error relative to the target. Thus, $\epsilon$ has no direct effect on the width or depth of the GNN.
>
>     (iii) GNN depth & width: Barcelo et al. show that the depth and width required to learn a function over graphs depends on the quantifier depth of the target logical sentence. The depth is the same in our case, and as for the width, we need to choose "the dimension of the state vectors in such a way that it is greater than $c ⋅ n^2$  and is at least as large as the dimension of the state vectors of $\mathcal{N}_f$" (Proof of Lemma A.1).
>
> 2. As stated in point 1, the GNN width and depth required for learning a function rely on the complexity of the target sentence and hence the target function. We distinguish Lemma A.1 (Boolean functions) and Theorem 4.1 (general functions) in our response:
>
>     **Lemma A.1.** In Lemma A.4, we show that any Boolean function over the set of graphs can be represented as a formula in$\ \text{C}^2$, and thus that MPNNs with RF can learn any Boolean function over graphs. As the reviewer correctly points, the construction can potentially yield an exponentially large sentence, but this is not necessarily the case, since this depends tightly on the target function h. There are two main reasons for this: (a) the target function h may require only a sentence that uses at most polynomially many disjuncts in our construction (i.e., the set H may be of polynomial size), and (b) the target function h can be captured by a logically equivalent (but much shorter) sentence than the one given in our construction. That is, our proof relies on the existence of a $\text{C}^2$ formula, but does not make any specific assumptions based on the formula derived in Lemma A.4. Hence, our construction is very adaptive and tightly linked to the descriptive complexity of the function we want to approximate, and it is easy to see that much more efficient constructions can be given for a more restricted class of functions. As we state in Remark 1, this deserves a more thorough investigation, and an important direction for future work is to understand, e.g., the class of functions that can be approximated via poly sized GNN.
>
>     **Theorem 4.1.** While lifting the result from Boolean functions to the general case, we rely on exponentially many dimensions, and this blow-up seems unavoidable in the general case, where we put no restrictions on the class of functions. The adaptive nature of our constructions  also has implications on Theorem 4.1, yielding better bounds. For instance, if the target function has a compact range, then we can cover the domain with a bounded number of points,  which only depends on the desired error epsilon.
>
>     We have now explicitly introduced the bound on width, and provided explanations of all these points, following Theorem 4.1 (Page 5).
> 3. As described in point 2, the construction aims to show the existence of a $\text{C}^2$ sentence capturing the target function. It is possible, and quite plausible, to characterize the target function in terms of more succinct sentences, once we restrict our attention to a less general class of functions. This subtlety is the reason why the overall idea is not based on memorization of individual graphs, but rather on the complexity of the sentences being learned.
>
> 4. Sato et al. investigate partial RF on real-world datasets, and show that GNNs supplemented with partial RFs perform comparably, if not marginally better, on standard datasets MUTAG, NCI1, and PROTEINS. This means that partial RF does not seem to hurt performance on real-world datasets. This confirms the intuition about the impact of RF we present in the paper, and further justifies the need for datasets EXP and CEXP.
>
> **CEXP/EXP**: We share the datasets in the supplementary material.
>
> [1] Barcelo' et al. "The Logical Expressiveness of Graph Neural Networks", ICLR 2020.
> [2] Sato et al. "Random Features Strengthen Graph Neural Networks.", arXiv 2002.03155

---

### Public Comment · ~Christopher_Morris1 · 2020-11-10
**Additional related work**

Interesting work! You might want to include https://www.ijcai.org/Proceedings/2020/294 in your related work.

---

> ### Author Response · Authors · 2020-11-12
> **Author Response**
>
> Thank you for taking an interest in our work, and for bringing this paper to our attention. We have now included this paper in our related work (Page 4), and discuss its contributions and compare with our own work.

---

> ### Public Comment · ~Pan_Li2 · 2020-11-16
> **Regarding the results on 1-2-3-GCN-L vs 3-GCN**
>
> I saw in the Table 1 of this work. The performance of 1-2-3-GCN-L (sparse version) is much worse than 3-GCN (dense version). As an author of https://grlplus.github.io/papers/80.pdf, which has the theoretical support to say that these two versions should not have any difference in representation power, do you have any comments on this result?

---

> > ### Author Response · Authors · 2020-11-17
> > **Author Response**
> >
> > We understand that this question is tailored to Christopher Morris, but we can shortly comment to clarify this: It is important to note that 1-2-3-GCN-L corresponds to $\delta-k-$LWL in the paper you cite, and not to $\delta-k-$LWL+. The latter is shown to have equivalent power to $\delta-k-$WL (Theorem 1), not the former. We use the former model 1-2-3-GCN-L (from [1]). Furthermore, in our case, our data falls exactly into the set of graphs where this local relaxation ultimately costs the model the ability to distinguish between non-isomorphic pairs. Thus, the 50% is not “much worse” in a quantitative sense, but rather in a qualitative sense, as the model itself cannot learn to distinguish between our data instances. You can refer to the third paragraph of Section 6.1 for a more detailed explanation of this result.
> >
> > [1] Christopher Morris, Martin Ritzert, Matthias Fey, William L. Hamilton, Jan Eric Lenssen, Gaurav
> > Rattan, and Martin Grohe. "Weisfeiler and Leman go neural: Higher-order graph neural networks". AAAI 2019

---

### Public Comment · ~Pan_Li2 · 2020-11-16
**Some relevant works and questions**

Two relevant works

"To date, higher-order invariant and equivariant networks are the only models with known universality results, but these results are practically hindered by prohibitive computational complexity."

*This argument is not quite right due to the recent work https://proceedings.neurips.cc/paper/2020/hash/2f73168bf3656f697507752ec592c437-Abstract.html

*Injecting randomness may indeed improve the representation power. Actually, there was a previous work that also actually put some relevant theory on this line: https://openreview.net/forum?id=SJxzFySKwH.

Overall, I think this is a very nice work! The results match the expectation, where injecting randomness may slow down the convergence but provide may representation power. I have another two questions that the authors may have more experiences:

1. One may inject randomness during the training. When performing inference, does one still injective randomness multiple times before making the final decision? If so, how many times of injection does one need to make the final inference?
2. The synthetic datasets constructed here are useful to evaluate the theory. However, how does RNI work for practical applications? Have the authors tried to use this approach in some more practical tasks？

---

> ### Author Response · Authors · 2020-11-17
> **Author Response**
>
> Thanks for bringing this to our attention.
>
> >  "This argument is not quite right due to the recent work https://proceedings.neurips.cc/paper/2020/hash/2f73168bf3656f697507752ec592c437-Abstract.html"
>
>    a) This work was published after our ICLR submission. Nonetheless, we are currently looking at recent works published in this time, and will make all necessary additions to our related work in the final version of our paper.
>    b) We acknowledge that this work also provides expressiveness improvements on standard GNNs. However, to the best of our understanding, this work does not provide a general universality result, unlike higher-order models. In fact, it states that its proposed models (DE-1) are themselves bound by WLGNNs over distance-regular graphs, which implies that this model cannot be universal. Only the special case of regular graphs is treated, and specific examples for DE-GNN-k, k>=2, are provided. Hence, the results of this work do not contradict with the quoted statement, as it is still the case with them that no known [universal] models are practically feasible.
>    c) If anything was misunderstood in this work on our part, we are happy to receive clarifications, and will make any necessary adjustments to the paper accordingly.
>
> > "Injecting randomness may indeed improve the representation power. Actually, there was a previous work that also actually put some relevant theory on this line: https://openreview.net/forum?id=SJxzFySKwH"
>
>    This result is indeed related, and we will include this in the next version of our work.
>
> > "Overall, I think this is a very nice work! The results match the expectation, where injecting randomness may slow down the convergence but provide may representation power"
>
>    We are pleased you find our work interesting. Indeed, these expectations were what motivated our study and results.
>
> **Responses to Questions:**
>
> 1. In our experiments, we inject randomness at every training/evaluation run, and compute a unique result from this randomization. That is, we simply add random features and compute the output in a standard fashion, without averaging across multiple different randomizations. We have experimented with such an averaging scheme when developing this work, but this did not provide any meaningful improvements: As one would expect, it led to slightly faster (but still relatively slow) convergence, as the probability of success naturally increases, but did not improve overall performance in any other respect.
> 2. Such a study has been conducted by Sato et al. [1] on three standard datasets, namely Proteins, NCI1 and MUTAG. There, the experiments run an RNI model analogous to ours and show that the model maintains the performance levels of deterministic GNNs, and even improves it marginally in some cases. Generally speaking, RNI is especially useful on challenging instances requiring higher expressive power, where GNNs otherwise fail and, that's the reason we designed dedicated datasets. Please also refer to our responses to the reviewers for more details.
>
> [1] Sato et al. "Random Features Strengthen Graph Neural Networks.", arXiv 2002.03155

---

### Public Comment · ~Pan_Li2 · 2020-11-17
**Some questions regarding the proof**

I read through the proof and have the following question. I hope to get some pointers from the authors.

I am not quite sure why Lemma A.5 is needed. I must have missed anything so please correct me: The goal of Lemma A.5 seems to say that random features lead to identifiable graphs with high probability. Actually, by randomly sampling features from [0,1] uniformly, with a finite number of samples n, with almost 1 probability, any two samples are different. This already makes all nodes distinguishable and gives identifiable graphs. Combining this result with Barcelo's result can give the conclusion directly.

I appreciate any feedback from the authors. Many thanks in advance!

Okay, I roughly understand. The real valued features have to be transformed into binary vectors to satisfy the condition of Barcelo's result. Many thanks! Nice proof!

---

> ### Author Response · Authors · 2020-11-17
> **Author Response**
>
> Thank you for your interest in this work, and we are pleased you enjoyed it. Indeed, Lemma A.5 is a technical lemma that enables us to achieve the 1-hot encoding of unary relations required for applying Barcelo et al's result in the proof of Lemma A.1.

---

### Author Response · Authors · 2020-11-23
**Author Summary**

We thank the reviewers and public commenters for their constructive feedback throughout this period. For greater convenience, we summarize and reiterate the main contributions of this work, and changes made thus far:

- **(Universality result)** The paper establishes, using an argument based on descriptive complexity, that GNNs enhanced with RNI (even with one random dimension) are universal. In addition to showing universality, this argument proves that the depth and size needed for the GNN entirely correlates with the complexity of the logical representation of the target representation, following on the result of Barcelo et al., and thus establishes a more grounded and principled study method of computational requirements for GNNs to learn classes of functions.  Our result substantially improves existing knowledge and results on the power of (partial) RNI, and also opens up new and interesting future directions for graph representation learning (e.g., is it possible to capture all polynomial functions via efficient constructions, based on existing findings in descriptive complexity?). Following reviewer feedback, we made these points clearer in the updated version of the paper, and provided more proof intuitions following the theorem statement.


- **(Universal and Permutation-invariant)** Our universality result preserves permutation-invariance of GNNs in expectation, a very desirable property. RNI introduces continuous node features which are re-sampled at every iteration of training or evaluation. These features, by construction, vary around a mean value and, in expectation, this mean value is what the GNN will rely on for predictions. However, the variability between different samples, and in particular that of a random sample relative to its mean, enables graph discrimination and improves expressiveness. Hence, in expectation, RNI preserves invariance, as all samples ultimately converge on average to a single value that can be sampled, whereas variance is leveraged for increased expressiveness. We clarified this further in the highlighted part of the paragraph preceding Theorem 4.1 (Page 4).


- **(Empirical evaluation against higher-order models)**  The paper conducts a thorough and dedicated empirical analysis to evaluate RNI, particularly partial RNI, and compare with more powerful higher-order GNN models. In this evaluation, it establishes that even small amounts of RNI are sufficient to achieve sufficient expressiveness for GNNs, and enable competitive performance against higher-order models at a much smaller computational cost. We also show, somewhat reassuringly, that GNNs with RNI converge much slower than higher-order models, which reflects the difficulty of the learning task in the GNN-RNI setting relative to the latter. We are not aware of any existing work that reports such empirical findings. Following feedback, we clarified the nature of our RNI, to make sure that RNI is explicitly described as being over feature vectors, and not over a subset of nodes.


- **(Novel datasets)** The paper introduces two carefully designed datasets, namely EXP and CEXP, which explicitly and unambiguously evaluate the expressive power of GNNs over a challenging problem setting, SAT. EXP is shown to strictly require 2-WL expressive power, and is thus beyond the ability of standard MPNNs, and even certain relaxed higher-order models. Furthermore, SAT cannot be solved over EXP/CEXP by detecting the presence or absence of a fixed set of structures, as these are shared across both positive and negative instances. Finally, CEXP carefully manipulates EXP data instances to preserve all these properties, while enabling partial 1-WL discrimination, so as to effectively appraise the impact of RNI on ‘simpler’ data jointly with the harder EXP-based instance, where RNI for MPNNs is strictly required. These datasets are provably challenging and we hope that they will be used more broadly in the community, and following feedback, we have made them available.


- **(Practicality)** Our result, supplemented with empirical evidence over EXP and CEXP corroborating the usefulness of partial RNI, establishes a practically viable solution for improving GNN expressiveness over hard graph instances, while not affecting existing performance on more standard data (also see real-world dataset results in Sato et al.). That is, we consider a well-known practice of RNI, and show that it is indeed a very viable option, especially for datasets that require more expressive power.

We hope our responses and this summary clarify all your concerns, and are happy to address any further questions you may have.

---

### Comment · ~Muhammet_Balcilar1 · 2021-01-19
**PPGN Results on EXP Dataset**

Thanks for this great work. I like to read the paper. Especially many thanks for the dataset that probably you spent a lot of time preparing it. Indeed it would be a great idea to test the empirical expressive power of GNN by this task.

However, I myself implemented Maron's 2-FWL equivalent network which was called PPGN in the paper. I do not think that the failure of the PPGN can be explained by not to learn required PMP function in the readout layer. According to my test, seems it can learn %100 and very fast. I am quite confident that there would be some implementation error.

I strongly think that we will see this paper in some high impact journal, conference soon.

---

> ### Author Response · Authors · 2021-01-26
> **Author Response**
>
> Thanks for your kind words, and for your interest. We are glad to see you have reproduced our results. Regarding PPGN, we used the original repository for the PPGN paper, available at https://github.com/hadarser/ProvablyPowerfulGraphNetworks, and thus did not implement that model ourselves. In fact, we simply converted our dataset into the necessary format for that implementation (also shared in our dataset files), and ran our experiments. We will investigate this more closely, and update this result accordingly.
>
> Designing the dataset was indeed a very challenging endeavour, as it had to simultaneously satisfy so many different criteria, but fortunately this effort bore fruit. We also think this dataset, along with our theoretical results, provide novel insights and understanding about graph neural networks and their expressive power.

---

### Author Response · Authors · 2021-06-14
**IJCAI’21 Camera-Ready version, Codebase, and datasets**

We would like to announce that our work has been accepted for publication at IJCAI 2021. The latest, revised version of the work can be found on http://www.arxiv.org/abs/2010.01179, and its codebase and datasets are available on http://www.github.com/ralphabb/GNN-RNI. Thanks to the reviewers and public commenters for their feedback!

---

### Decision · Program_Chairs · 2021-01-07
**Final Decision**

**Decision:**

Reject

**Comment:**

In this paper, the authors show the effect of RNI on the expressive power of GNN for the first time, where the RNI was initially proposed in Sato et al. 2020. Overall, I like the idea of random node initialization because it is simple, effective, and theoretically well-founded. The key concern was that the novelty over the Sato's paper and the reviewers were still not convinced by the response. Therefore, the paper is still below the acceptance threshold.  I strongly encourage authors to revise the paper based on the reviewer's comments and resubmit it to a future venue.